# DERD-Net: Learning Depth from Event-based Ray Densities

**Diego Hitzges**[*1]    **Suman Ghosh**[*1]    **Guillermo Gallego**[1,2]

[1]Technische Universität Berlin, Einstein Center Digital Future, Robotics Institute Germany
[2]Science of Intelligence Excellence Cluster, Germany

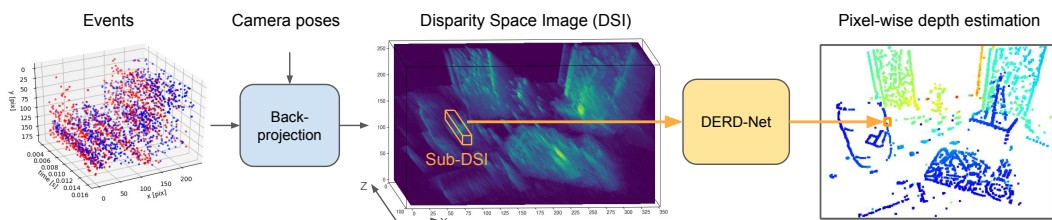

Figure 1: *Overview*. We present a deep-learning–based method to predict depth from event-ray densities (Disparity Space Images –DSIs) obtained by back-projecting events using camera poses. Our deep neural network, DERD-Net, operates in parallel on local volumetric neighborhoods of the DSI data, called Sub-DSIs (in orange).

## Abstract

Event cameras offer a promising avenue for multi-view stereo depth estimation and Simultaneous Localization And Mapping (SLAM) due to their ability to detect blur-free 3D edges at high-speed and over broad illumination conditions. However, traditional deep learning frameworks designed for conventional cameras struggle with the asynchronous, stream-like nature of event data, as their architectures are optimized for discrete, image-like inputs. We propose a scalable, flexible and adaptable framework for pixel-wise depth estimation with event cameras in both monocular and stereo setups. The 3D scene structure is encoded into disparity space images (DSIs), representing spatial densities of rays obtained by back-projecting events into space via known camera poses. Our neural network processes local subregions of the DSIs combining 3D convolutions and a recurrent structure to recognize valuable patterns for depth prediction. Local processing enables fast inference with full parallelization and ensures constant ultra-low model complexity and memory costs, regardless of camera resolution. Experiments on standard benchmarks (MVSEC and DSEC datasets) demonstrate unprecedented effectiveness: (i) using purely monocular data, our method achieves comparable results to existing *stereo* methods; (ii) when applied to stereo data, it strongly outperforms all state-of-the-art (SOTA) approaches, reducing the mean absolute error by at least 42%; (iii) our method also allows for increases in depth completeness by more than 3-fold while still yielding a reduction in median absolute error of at least 30%. Given its remarkable performance and effective processing of event-data, our framework holds strong potential to become a standard approach for using deep learning for event-based depth estimation and SLAM. Project page: https://github.com/tub-rip/DERD-Net

---

[*]Equal contribution

39th Conference on Neural Information Processing Systems (NeurIPS 2025).

# 1 Introduction

Depth estimation is a fundamental task in computer vision, with key applications in areas such as robotics, autonomous driving, and augmented reality. Traditional stereo vision techniques rely on synchronized cameras to capture images and infer depth by finding correspondences between them. However, these methods often struggle in low-light and fast-motion conditions. Moreover, conventional cameras produce large amounts of redundant data by capturing entire images at fixed intervals, leading to inefficiencies in both data storage and processing.

Unlike conventional cameras, event cameras operate asynchronously, detecting per-pixel brightness changes (called "events") [1–3]. This provides high temporal resolution and robustness to motion blur, making them well suited for dynamic scenes. Their sparse output enables efficient processing of only relevant areas, making them ideal for real-time tasks such as visual odometry (VO) / SLAM.

Harnessing deep learning for depth estimation with event cameras has the potential to transform such applications. However, adapting neural networks to event data remains challenging due to its asynchronous stream-like nature. Moreover, the scarcity of event camera datasets with ground truth depth [4, 5] results in limited training data, which can lead to overfitting [6]. While simulating data is one way to address this issue, generalizing models trained on simulation to real-world scenarios is far from trivial due to differences in data distribution [7–10]. Thus, instead of directly processing events (which is prone to overfitting and extensive training time) we rely on intermediate event representations informed by the scene geometry.

One promising approach for 3D reconstruction is back-projecting events as rays into space and capturing their intersection densities as disparity space images (DSIs) [11]. DSIs from two or more cameras can be fused, eliminating the need for event synchronization between cameras. This reduces complexity and allows for more robust depth estimation. The Multi-Camera Event-based Multi-View Stereo (MC-EMVS) method [12] recently produced state-of-the-art (SOTA) results, outperforming other techniques in depth benchmarks across several metrics. To obtain pixel-wise depth estimates from a DSI, MC-EMVS selects the disparity level with the highest ray count, effectively using an argmax operation. Ray counting is used as a proxy for finding 3D edges where the rays intersect. A selective threshold filter is applied to predict depth only for pixels with sufficient ray counts.

While straightforward, this approach does not fully utilize the potential of DSIs. It is susceptible to noise and cannot effectively extract cues from surrounding pixels and more complex patterns across disparity levels. Consequently, it leads to fewer pixels obtaining depth estimation and less accurate depth predictions than the DSI might potentially allow for. Therefore, we need more effective approaches for extracting depth from DSIs, that are more reliable, accurate, and produce more depth estimates, by adequately recognizing complex ray intersection patterns.

**Our Contribution**. We propose a novel deep learning framework for event-based depth estimation that is optimized for SLAM scenarios and addresses the aforementioned limitations. An overview is illustrated in Fig. 1. Our approach estimates pixel-wise depth from a DSI using a neural network with 3D convolutions and a recurrent structure. The framework is directly applicable to both monocular and stereo settings. Our key contributions include:

- **Learning-based Local Processing**: For each selected pixel, a small local subregion of the DSI (Sub-DSI) is used as input to the neural network. This novel design leverages the inherent sparsity of event data and DSIs to efficiently process and produce only relevant information for sparse depth-related tasks. (Sec. 3).

- **Enhanced Data Utilization**: Our model captures complex patterns within the DSI, increasing depth prediction accuracy and the number of pixels for which depth can be reliably estimated. Limiting the input to small local subregions around selected pixels enhances generalization by preventing the network from overfitting to dynamics specific to the training scenes and augmenting the available training set, as each Sub-DSI serves as an individual data instance.

- **Efficiency and Scalability**: By adopting our Sub-DSI approach, we obtain small independent inputs of fixed size. This enables full parallelization and provides an ultra-light network that has the ability to handle *any camera resolution* with constant very short inference time. Our network architecture allows the processing of DSIs of variable depth resolution. (Sec. 3.2).

- **Comprehensive Experiments**: We evaluate our model on both monocular and stereo data from the standard datasets MVSEC [4] and DSEC [13] using cross-validation. It outperforms the

state of the art by a large margin on ten figures of merit. Even using monocular data, our model achieves performance comparable to SOTA methods that require stereo data (Sec. 4). We show downstream applicability and robustness to imperfect (noisy) camera poses.

To the best of our knowledge, our work is the first learning-based multi-view stereo method to (i) use camera poses along with events as input, which is crucial for accurate depth estimation over long intervals (as used in SLAM [14, 15]); (ii) demonstrate successful and robust depth prediction on real-world event data from DSIs; (iii) report good generalization on all *three* MVSEC *indoor flying* sequences [5], even when compared to multi-modal methods that combine stereo intensity frames with events. We provide code, trained models and video results for clarity and reproducibility.

## 2 Related Work

**Stereo depth estimation** with event cameras has been a captivating problem since the invention of the first event camera by Mahowald and Mead in the 1990s [3, 5, 16] due to their potential for high temporal resolution and robustness to motion blur. Recent approaches have addressed stereo event-based 3D reconstruction for VO and SLAM [14, 17–22]. These methods assume a static world and known camera motion, using this information to assimilate events over longer time intervals, thereby increasing parallax and producing more accurate semi-dense depth maps. A comprehensive review is provided in [5].

MC-EMVS [12] introduced a novel stereo approach for depth estimation which does not require explicit data association, using DSIs generated from stereo events cameras. By leveraging the sparsity of events and fusing back-projected rays, they outperformed the event-matching–based solution of [19] and thereby achieved SOTA results in 3D reconstruction and VO [14]. Evidently, this DSI-based 3D reconstruction is robust to imperfect poses estimated using an event camera tracking method. We advance this approach by employing a compact neural network specialized to derive explicit depth from the DSIs, creating a standardized and effective framework for processing event-based data in deep learning applications that does not rely on event simultaneity or matching.

**Deep Learning for depth estimation from event data**. Deep learning has significantly advanced depth estimation in traditional monocular and stereo camera setups, achieving remarkable results [23–25]. However, its application to event camera data remains relatively limited due to the sparse asynchronous nature of event streams, which require specialized frameworks [5]. For example, in monocular vision, [26] uses synthetic data on a recurrent network to capture temporal information from grid-like event inputs. Yet, mismatches between synthetic and real data degrade performance [27], and monocular depth estimation from events is an ill-posed problem, making high accuracy challenging to achieve with this learning-based framework [28].

For stereo depth estimation, [6, 29] present two pioneering studies. Specifically, DDES [6] introduced the first deep-learning–based supervised stereo-matching method, while [29] proposed the first unsupervised learning framework. Both methods use First-In First-Out queues to store events at each position, allowing for concurrent time and polarity reservation. Nevertheless, high event rates lead to greater processing demands and, consequently, increased model complexity and memory requirements, limiting the use of visual cues from both event cameras. Our method overcomes these challenges by maintaining constant input dimensions defined by the size of the DSI subregion around a selected pixel, regardless of the event rate. It thereby provides a significant advancement in applying deep learning to event-based depth estimation on real-world data.

## 3 Methodology

In this section, we present our supervised-learning–based approach and the related framework in detail, for which a general overview is provided in Fig. 1. We describe the preprocessing of the data, the architecture of our network (shown in Fig. 2) and the training and inference procedures.

### 3.1 Framework

As an event camera with $W \times H$ pixels moves through a scene, it triggers events $e_k = (x_k, y_k, t_k, \pm_k)$ and produces a near continuous-time stream of data $\mathcal{E} = \{e_k\}$. Following [11, 12], this stream is

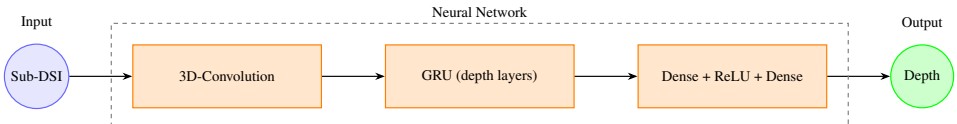

Figure 2: Network Architecture. The parameters of the network's modules are specified in Tab. 1.

Table 1: Details of network layers. K, P, and S stand for kernel-size, padding and stride.

| Layer | Dimensions | Details |
|---|---|---|
| Sub-DSI (Input) | $100 \times 1 \times 7 \times 7$ | Depth $\times$ Channels $\times$ Width $\times$ Height |
| 3D-Convolution | $50 \times 4 \times 5 \times 5$ | K = (3,3,3); P = (1,0,0); S = (2,1,1) |
| ReLU + Flatten | $50 \times (4 \cdot 5 \cdot 5)$ | Flattens channels and frame |
| GRU | $1 \times 100$ | Selects final hidden state $h_{50}$ |
| Dense + ReLU | $100$ | Maintains dimension |
| Dense (Output) | $1$ or $3 \times 3$ | Outputs depth value(s) |

sliced into time intervals. For every time interval, a DSI is created and associated with a camera viewpoint (called Reference Viewpoint) as follows: given camera poses, all events $e_k$ in the interval are back-projected into 3D space by casting rays from the moving camera optical center through the corresponding pixel $(x_k, y_k)$. The depth axis is discretized into $D$ levels, equidistant in inverse linear space, resulting in a 3D DSI of size $D \times W \times H$ voxels, whose values represent the number of rays passing through each region (i.e., voxel) of space (as shown in Fig. 1). Although input poses are required for building a DSI, they can be obtained from tracking methods or dead-reckoning [12,14,30].

We consider a stereo setup with two synchronized cameras providing two perspectives of the same scene. By leveraging parallax, this configuration enhances depth perception and allows for more accurate 3D reconstruction. For each interval, we construct two DSIs (one for each camera) and fuse them by applying voxel-wise metrics (e.g., harmonic mean) as described in [12]. To compare performance, we also apply our approach to the monocular data of the left camera only.

Since DSIs are typically large and sparse, depth is estimated only for pixels with sufficient information. A confidence map is generated by projecting the DSI onto a 2D grid of size $W \times H$, where each pixel's value represents the maximum ray density among all depth levels [12] (called pixel selection map in Fig. 3). An adaptive Gaussian threshold (AGT) filter is then applied to this grid to select the pixels $\{p_1, \ldots, p_n\}$ with a sufficient maximum ray density for reliable depth estimation. For each selected pixel $p_i = (x_i, y_i)$, a surrounding subregion $\tilde{S}_i$ is extracted from the DSI, including the ray counts of all pixels within L1 radii of $r_W, r_H$:

$$\tilde{S}_i \doteq \text{DSI}[\,:\,,\; x_i - r_W : x_i + r_W,\; y_i - r_H : y_i + r_H].$$

Each subregion $\tilde{S}_i$ is then normalized individually,

$$S_i \doteq \tilde{S}_i \,/\, \max(\tilde{S}_i) \in [0,1]^{D \times (2r_W + 1) \times (2r_H + 1)}, \tag{1}$$

and serves as input to our neural network. Using only small subregions of the DSIs as inputs leads to an efficient and compact architecture operating independently of camera resolution. Furthermore, it reduces the risk of overfitting by encouraging the model to learn generalizable patterns of *ray intersections* within the Sub-DSIs instead of memorizing semantics and dynamics specific to the training scene. Such localized input processing is possible because DSIs consolidate sparse events into a structured format that preserves geometric information within small spatial regions.

The depth estimates $z_1, \ldots, z_n$ are computed in parallel. Since the amount of selected pixels can be controlled by the AGT filter and the dimensions of the Sub-DSIs are fixed, we achieve constant low model complexity and memory costs, regardless of the number of triggered events.

### 3.2 Network Architecture

The architecture of the neural network is illustrated in Fig. 2, with the dimensions of each layer listed in Tab. 1. The network receives the normalized Sub-DSI (1) as input. The objective is to capture local geometrical patterns in the Sub-DSI to extract more relevant depth information than the SOTA argmax approach used in [11, 12]. Since established networks like U-Net [31] often include strong spatial

Table 2: Hyperparameters.

| Dataset | Dataset Details | | | DSI Parameters | | | | | Gauss. Filter | | Training Process | | | | |
|---|---|---|---|---|---|---|---|---|---|---|---|---|---|---|---|
| | Sequences | Resolution | LiDAR $\Delta t$ | Span | $z_{min}$ | $z_{max}$ | D | Sub-DSI | Window | C | Batch | Optimizer | LR | LF | Epochs |
| MVSEC | Indoor flying | $346 \times 260$ px | 50 ms | 1 s | 1 m | 6.5 m | 100 | $7\times7$ | $9\times9$ | $-10$ | 64 | AdamW | $10^{-3}$ | MAE | 3 |
| DSEC | Zurich04a | $640 \times 480$ px | 100 ms | 0.2 s | 4 m | 50 m | 100 | $7\times7$ | $9\times9$ | $-2$ | 64 | AdamW | $10^{-3}$ | MAE | 3 |

compression and Transformers tend to impose high data demands for reliable generalization [32], we tailor an ultra-lightweight custom architecture that shares design elements with FireNet [33], adapted for very small frame sizes yet variable depth dimensions. Similarly to RAFT-Stereo [34], we adopt convolutions and a Gated Recurrent Unit (GRU) to efficiently handle different depth resolutions, enabling customization of the desired depth precision without modifying architecture.

First, to capture local patterns, further reduce input size and avoid overfitting, we apply a 3D convolutional filter (3D-Conv) with a ReLU activation, a kernel size of $3 \times 3 \times 3$ and no padding in the spatial dimensions (width and height). For the depth dimension, we set a padding of 1 and a stride of 2 to halve the number of depth layers. We use 4 output channels to capture different patterns simultaneously, resulting in the convolved version of the Sub-DSI (1):

$$S_i^* = \text{3D-Conv}(S_i) \in \mathbb{R}^{\frac{D}{2} \times 4 \times (2r_W - 1) \times (2r_H - 1)}. \tag{2}$$

To create the DSIs, rays were cast from the representative camera location into space, passing sequentially through the different depth levels. Since mapping precision requirement may change from scene to scene, we need to deal with variable depth resolution of the DSI. To efficiently and flexibly model this interdependence of consecutive depth layers for a *variable D*, the convolved depth layers are flattened and successively fed into a GRU [35]. The recurrent structure of the GRU allows us to maintain a constant ultra-low count of only 70k parameters in total. It iteratively embeds each depth layer's information within the context of the previous layers, producing hidden state representations $h_1, \ldots, h_{D/2}$. We then proceed with the final hidden state $h_{D/2}$, which condenses the relevant information from all depth layers along the depth axis:

$$h_{\frac{D}{2}} = \text{GRU}(S_i^*) \in \mathbb{R}^{4 \cdot (2r_W - 1) \cdot (2r_H - 1)}. \tag{3}$$

Finally, a dense layer that preserves the dimension of $h_{D/2}$ with ReLU activation and a subsequent output dense layer are applied to process the hidden state. We introduce two versions of the network for custom modification of depth estimation density. In the single-pixel version, the network predicts the normalized depth for the selected pixel $z_i \in [0, 1]$, while in the multi-pixel version, the output is a $3 \times 3$ grid $Z_i \in [0, 1]^{3 \times 3}$, representing the normalized depth predictions of the central pixel and its 8 neighbors. Finally, normalized depth is converted into actual depth by mapping $[0, 1]$ to $[z_{min}, z_{max}]$.

### 3.3 Training and Inference

As supervised loss function for training the neural network model we use the *mean absolute error (MAE)*, with given ground truth depth (this is the case of standard real-world datasets used, such as MVSEC and DSEC – see Sec. 4). To reduce training time and further improve generalization, we additionally employ ensemble learning (EL) [36, 37]. For training, we initialize two identical but independent instances of our neural network with different random weights. The training set is split into two disjoint subsets, enabling parallel training. During testing or inference, each Sub-DSI $S_i$ is processed simultaneously by both networks, and the final depth estimation is obtained by averaging the individual predictions. This helps reduce variance in the predictions, leading to more stable and accurate results. We also present results without EL in Tabs. 11 to 14 in the Appendix Sec. B.

## 4 Experiments

In this section, we evaluate the performance and reliability of the proposed depth estimation approach. Following prior protocols, we conduct experiments on the MVSEC [4] and the DSEC [13] datasets. Ground truth (GT) depth, captured at fixed intervals using LiDAR sensors, serves as reference locations for constructing the respective DSIs over a defined time span. Pixel selection for depth estimation is based on an AGT filter, where the window size determines the surrounding pixel count considered, and a constant $C$ is subtracted from the observed ray count.

Table 3: Summarized quantitative comparison of the proposed methods with the state of the art. *MVSEC indoor_flying* and *DSEC Zurich_City_04_a*. The full comparison over ten metrics is in the Appendix Sec. B.

| | Method | | MVSEC | | | | DSEC | | | |
|---|---|---|---|---|---|---|---|---|---|---|
| | Algorithm | Modality | Mean Err [cm] ↓ | Median Err [cm] ↓ | bad-pix [%] ↓ | #Points [million] ↑ | Mean Err [m] ↓ | Median Err [m] ↓ | bad-pix [%] ↓ | #Points [million] ↑ |
| SOTA | EMVS [11] | monocular + $F_{\text{orig}}$ | 33.78 | 14.35 | 3.84 | 1.27 | 5.64 | 2.52 | 13.68 | 1.31 |
| | EMVS [11] | monocular + $F_{\text{denser}}$ | 50.32 | 20.81 | 11.46 | 4.15 | 7.01 | 3.56 | 24.33 | 6.09 |
| | ESVO [19] | stereo | 22.70 | 9.83 | 2.83 | 1.56 | 3.93 | 1.62 | 10.54 | 9.40 |
| | SGM [38] | stereo | 35.42 | 12.35 | 6.39 | **14.46** | 6.74 | 1.58 | 15.25 | 8.30 |
| | GTS [39] | stereo | 389.00 | 45.43 | 38.45 | 0.06 | 26.24 | 1.62 | 32.56 | 0.11 |
| | MC-EMVS [12] | stereo + $F_{\text{orig}}$ | 20.07 | 9.53 | 1.35 | 0.81 | 3.27 | 0.90 | 10.75 | 1.25 |
| | MC-EMVS [12] | stereo + $F_{\text{denser}}$ | 28.38 | 12.38 | 3.26 | 2.77 | 4.76 | 1.56 | 17.42 | 4.64 |
| | MC-EMVS [12] + MF | stereo + $F_{\text{orig}}$ | 20.64 | 9.72 | 1.43 | 3.00 | 3.51 | 0.96 | 11.81 | 3.83 |
| Ours | DERD-Net | monocular + $F_{\text{orig}}$ | 23.68 | 11.55 | 2.78 | 1.21 | 3.12 | 1.60 | 5.50 | 2.10 |
| | DERD-Net | monocular + $F_{\text{denser}}$ | 28.52 | 13.85 | 4.87 | 4.15 | 3.01 | 1.50 | 6.35 | 6.09 |
| | DERD-Net | stereo + $F_{\text{orig}}$ | **11.69** | **5.50** | **0.89** | 0.79 | 1.61 | **0.46** | 4.12 | 1.67 |
| | DERD-Net | stereo + $F_{\text{denser}}$ | 15.24 | 6.68 | 1.70 | 2.77 | 1.80 | 0.54 | 5.04 | 4.64 |
| | DERD-Net (multi-pixel) | stereo + $F_{\text{orig}}$ | 12.02 | 5.63 | 0.90 | 4.32 | **1.59** | 0.47 | **3.81** | 6.59 |
| | DERD-Net (multi-pixel) | stereo + $F_{\text{denser}}$ | 15.68 | 6.73 | 1.74 | 11.33 | 1.79 | 0.54 | 4.61 | **14.74** |

We investigate the impact of DSIs derived from both monocular and stereo settings during training and testing. Table 2 provides an overview of the key parameters used in the datasets and the training processes of the experiments. Abbreviations are: minimum depth ($z_{\min}$), maximum depth ($z_{\max}$), depth dimensions ($D$), filter window size (Window), subtractive constant ($C$), batch size (Batch), learning rate (LR), and loss function (LF).

### 4.1 Metrics

The performance of the networks is evaluated using *ten* standard metrics commonly employed in depth estimation tasks [12]. We calculate both mean and median errors between the estimated and GT depths, with median errors providing robustness against outliers. Additionally, we report the number of reconstructed points, reflecting the algorithm's ability to generate valid depth estimations, and the number of outliers (bad-pix [40]), representing the proportion of significant depth estimation errors. In the Appendix, we also compute the scale-invariant logarithmic error (SILog Err) to evaluate the error while considering scale, and the sum of absolute relative differences (AErrR) to assess the relative accuracy of the depth predictions. Finally, we report $\delta$-accuracy values, which indicate the percentage of points whose estimated depth falls within specified limits relative to GT [41].

### 4.2 Baseline Methods

We compare our approach against several SOTA methods that have been benchmarked on the task of *long-term* event-based depth estimation [5], thus evaluated under the same input conditions (events and camera poses) and output format (semi-dense depth maps) supportive of SLAM. In the absence of other deep stereo methods that learn from input camera poses, we also include comparisons against the SOTA instantaneous end-to-end learning-based stereo methods in Sec. 4.5 for completeness. We adopt the same train-test splits established in prior work and standard benchmarks [5].

The Generalized Time-Based Stereovision (GTS) method [39] utilizes a two-step process: first performing stereo matching based on a time-consistency score for each event, followed by depth estimation through triangulation. The Semi-Global Matching (SGM) method [38] is adapted for event-based data by generating time images and subsequently applying stereo matching, with the depth map being refined by masking it at the locations of recent events. Another method, Event-based Stereo Visual Odometry (ESVO) [19] (ESVO2 [42]), integrates depth estimates by employing Student-t filters, ensuring robust spatio-temporal consistency between stereo time image patches.

The two closest baseline methods for performance comparison of our method are EMVS for monocular vision [11] and MC-EMVS for stereo vision [12]. Both methods extract pixel-wise depth from DSIs by applying the argmax function. To ensure consistency and fairness, we benchmark the methods following the procedure established in prior works [5].

| Scene | Pixel selection map | MC-EMVS [12] | MC-EMVS [12] with $F_{denser}$ | DERD-Net (Ours) | Ground truth (GT) |
|---|---|---|---|---|---|

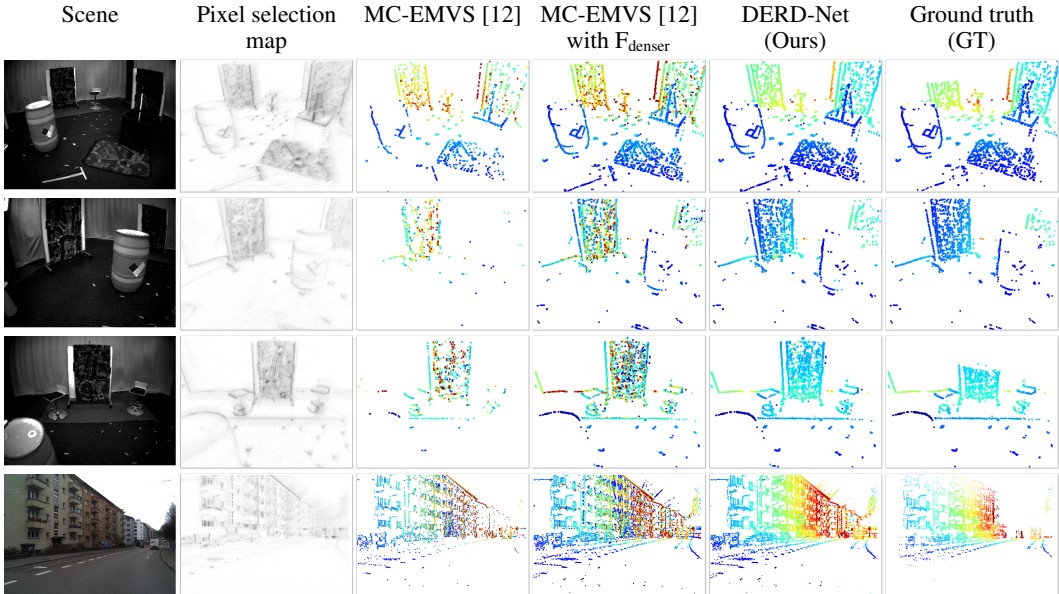

Figure 3: *Depth estimation*. Qualitative comparison of depth estimated using the MC-EMVS method [12], applying it to the new selected pixels $F_{denser}$ and our method DERD-Net, for the MVSEC *indoor_flying* [4] (top 3 rows) and DSEC *Zurich_City_04_a* (bottom row) sequences. Ground truth depth from LiDAR is masked by pixels with valid depth estimate. Our method estimates depth even at pixels with no GT depth. Depth maps are pseudo-colored, from blue (close) to red (far), in the range 1-6.5m for MVSEC and 4-50m for DSEC.

## 4.3 Experiments on MVSEC Dataset

This section describes the experiments conducted on the *indoor_flying* sequences 1,2,3 of the MVSEC dataset [4] to evaluate the performance of the proposed depth estimation method. Most stereo methods do not evaluate on *indoor_flying_4* (because of noisy events from the low-texture floor as the drone flies very low) and the driving sequences (because the stereo baseline is too small for the given depth range and low camera resolution). We employed three-fold cross-validation by utilizing two sequences for supervised training and reserving the remaining sequence for testing, repeating this process for all three possible combinations of sequences to ensure robustness in our evaluation.

**Two pixel-selection filter settings:** $F_{orig}$ **and** $F_{denser}$. We first trained the single-pixel version of our network on monocular DSIs to compare its performance to EMVS [11]. Subsequently, we retrained it on stereo DSIs fused via the harmonic mean and compared its performance to MC-EMVS [12]. These two baseline methods used an AGT filter $F_{orig}$ with a window of $5 \times 5$ px and a subtractive constant of $C_{orig} = -14$. Given that our network is designed to extract additional information from the geometrical patterns within the DSI, we hypothesized that it would still produce reliable depth estimates for pixels with lower confidence. To test this hypothesis, we used a larger filter window size of $9 \times 9$ px and a subtractive constant of $C_{denser} = -10$, resulting in a less strict filter $F_{denser}$, enabling depth estimation for more pixels. To ensure a representative comparison, we evaluated our networks as well as EMVS and MC-EMVS on both sets of pixels created by $F_{orig}$ and $F_{denser}$.

**Multipixel vs. Morphological Filter**. One apparent drawback of MC-EMVS is the limited number of pixels for which depth is estimated compared to other SOTA methods. To address this, [12] presented the option of adding a 4-neighbor morphological filter (MF), which dilates the depth estimation map to increase the number of depth-estimated pixels. We compare this with our framework's ability to further increase the number of depth-estimated pixels by training and evaluating the multi-pixel version of our network.

**Results**. The averaged results over all three sequences are displayed in the left half of Tab. 3 for all discussed modalities, while the individual results (including performance without EL) are detailed in the Appendix, in Tabs. 12 to 14. Notably, our network's performance converges after only 3 epochs

of training. This rapid convergence is particularly advantageous for future applications where the network might be trained on more heterogeneous datasets or retrained for specific scenarios.

**Monocular setup**. On the monocular DSIs filtered by $F_{\text{orig}}$, our single-pixel network achieves results comparable to those of *stereo* SOTA methods and significantly outperforms EMVS [11] by 30% in MAE. Remarkably, even on the 3.27 times larger set of pixels created by $F_{\text{denser}}$, it still achieves better scores than EMVS at $F_{\text{orig}}$ for all metrics, except bad-pix. Applying EMVS to the same expanded set of pixels leads to a 76% increase in MAE compared to our framework.

**Stereo setup**. Applying our single-pixel network to stereo DSIs filtered by $F_{\text{denser}}$ allows us to predict depth at significantly more pixels than any other method, except for the SGM method [38], while consistently surpassing all benchmarks across all metrics. The number of pixels increases by 242%, while the MAE and MedAE reduce by 24% and 30%, compared to MC-EMVS [12] with $F_{\text{orig}}$. The only exception is the bad-pix measure, where MC-EMVS performs slightly better. When both methods are compared on the same set of pixels, our approach yields a reduction in both MAE and MedAE of 42% for $F_{\text{orig}}$ and 46% for $F_{\text{denser}}$, respectively. Performance remains consistent when using the multi-pixel network, which increases the amount of depth-estimated pixels by a factor of 5.47 for $F_{\text{orig}}$ and 4.09 for $F_{\text{denser}}$ compared to its single-pixel version, indicating it to be a superior approach to the morphological filter of MC-EMVS, which only rises the number of points by a factor of 3.70 for $F_{\text{orig}}$. As a consequence, the multi-pixel version of our framework estimates depth for almost as many pixels as the SGM method while delivering new SOTA results.

**Qualitative comparison**. To further illustrate the effectiveness of our method, Fig. 3 compares depth maps generated by our single-pixel network against those produced by SOTA method MC-EMVS. Our network not only provides a denser depth estimation, which improves the recognition of contours, but also effectively eliminates the visible outliers produced by MC-EMVS. This improvement is evident when comparing our method to MC-EMVS applied both to the expanded and the original set of pixels, highlighting the robustness and superiority of our approach.

### 4.4 Experiments on DSEC Dataset

To assess the applicability of our network architecture to different data, we retrained and tested it on DSIs obtained from a stereo setting in the DSEC dataset [13]. This dataset presents unique challenges due to its outdoor driving scenarios, which differ significantly from the indoor environments of the MVSEC dataset, its higher spatial resolution ($640 \times 480$ px) and different noise characteristics (Prophesee camera vs. DAVIS346 camera). Moreover, straight driving sequences are especially challenging for event-based multi-view stereo due to the little motion parallax present in them.

**Setup.** We select the commonly used *Zurich_City_04_a* sequence to provide a focused in-depth evaluation. We split the sequence into two halves for training and testing. The DSIs were created by fusing the left and right DSIs via the harmonic mean. Analogous to Sec. 4.3, we use the original filter from MC-EMVS [12] $F_{\text{orig}}$ with a window size of $5 \times 5$ px and $C_{\text{orig}} = -4$, and a denser filter $F_{\text{denser}}$ with a window size of $9 \times 9$ px and $C_{\text{denser}} = -2$. First, the network was trained for 3 epochs on the DSIs of the first half of the sequence and tested on those of the second. The process was then reversed and each network was used to predict in its testing half of the data sequence.

**Results.** The results of these experiments are displayed on the right half of Tab. 3 and illustrated in Fig. 3. Our approach drastically outperforms every other method across all metrics, with our multi-pixel network achieving even slightly better performance than the single-pixel network. For $F_{\text{orig}}$, it reduces the MAE by 55% on a 1.72x higher number of pixels compared to MC-EMVS with a morphological filter. For $F_{\text{denser}}$, depth estimation density is increased by an additional factor of 2.24 while performance remains mostly stable, yielding a reduction in MAE of 62% compared to the argmax operation from MC-EMVS. Remarkably, even on purely monocular DSIs filtered by $F_{\text{denser}}$, our framework achieved superior performance to all benchmarked methods for every metric except MedAE. These results underscore the robustness and versatility of our approach, even in complex real-world outdoor scenes.

### 4.5 Robustness of DERD-Net compared to other deep-learning stereo methods

Since there are no comparable learning-based methods that use prior camera poses, Tab. 4 compares end-to-end learning-based stereo methods, which are "instantaneous" (do not take into account

camera poses) and output dense depth. In order to use their output for efficient VO/SLAM, we would need an extra step of extracting features (keypoints). Instead, DERD-Net's semi-dense depth maps help avoid unnecessary computation by outputting 3D edges for direct visual odometry, as in [14].

We use the same train-test splits established as the other learning-based methods [5]. While absolute accuracy is not directly comparable, evaluating the errors in the different splits relative to each other is informative about robustness: we observe that our method is the first one to generalize robustly across all three sequences (Tab. 4). No other method reports good generalization on "split 2" of MVSEC because of the difference in dynamic characteristics of events in training and testing on that split [6,43]. This is true even when compared to hybrid approaches, despite them also using stereo intensity frames ("2E+2F" input data modality). The observed robustness of our method to such shifts may be supported by the architectural choice of processing only small subregions as input (see Tab. 2 for Sub-DSI frame size), which encourages the model to learn generalizable patterns within the Sub-DSIs rather than memorizing global scene layout or dataset-specific context.

Table 4: Mean depth error [cm] of deep stereo methods on MVSEC indoor data. Values are collected from original sources.

| Method | Modality | Split 1 | Split 2 | Split 3 |
|---|---|---|---|---|
| DDES [6] | 2E | 16.7 | 29.4 | 27.8 |
| EIT-Net [43] | 2E | 14.2 | - | 19.4 |
| DTC-SPADE [44] | 2E | 13.5 | - | 17.1 |
| Liu et al [45] | 2E | 20 | 25 | 31 |
| StereoSpike [46] | 2E | 16.5 | - | 18.4 |
| ASNet [47] | 2E | 20.46 | 28.74 | 22.15 |
| Ghosh et al. [48] | 2E | 12.1 | - | 15.6 |
| Chen et al [49] | 2E | 13.9 | - | 14.6 |
| StereoFlow-Net [50] | 2E | 13 | - | 15 |
| EIS (ICCV 2021) | 2E + 2F | 13.74 | 18.43 | 22.36 |
| SCS-Net [51] | 2E + 2F | 11.4 | - | 13.5 |
| N. Uddin et al [29] | 2E + 2F | 19.7 | - | 26.4 |
| Zhao et al. [52] | 2E + 2F | 9.7 | - | 11.1 |
| **DERD-Net** | 2E | 11.69 | 11.11 | 12.28 |

## 4.6 Sensitivity Analyses

In this section we carry out experiments varying the settings in Tab. 2. Furthermore, we analyze the robustness of our method to noisy camera poses obtained from an event-based SLAM system.

**Sensitivity with respect to sub-DSI size**. Varying the horizontal and vertical extent of the Sub-DSIs has an impact on our method's performance. Our experiments show that the performance of DERD-Net can be improved by increasing the frame size of the Sub-DSIs, at the expense of increasing the network complexity (e.g., parameter count and computational cost). See Appendix Sec. A.1.

**Sensitivity with respect to DSI transformations**. We analyzed how DERD-Net behaves in the case of previously unseen but structurally similar environments, obtained by means of horizontal and vertical flips of the DSIs. Although its performance worsened slightly, it still outperformed all baseline methods. This demonstrates robustness to the aforementioned transformations. See Appendix Sec. A.2.

**Sensitivity with respect to noisy camera poses**. To assess the importance of having accurate camera poses during DSI construction, we test our framework using noisy poses with drift, mimicking real-world SLAM conditions. Instead of ground-truth (GT) poses from LiDAR-IMU odometry, we use poses estimated by the stereo event-based VO system ES-PTAM [14], which reports an Absolute Trajectory Error (ATE) of 131.62 cm over a 50 m-deep scene in the *DSEC Zurich_City_04_a* sequence. Running DERD-Net with these imperfect poses yields the results shown in the top rows of Tab. 5. The percentage values in parentheses denote the relative differences with respect to the performance obtained using ideal (GT) poses (Tab. 3). Remarkably, performance improved across all metrics (likely due to the slight reduction in the number of evaluated points of comparable magnitude), demonstrating strong robustness of DERD-Net to noisy poses obtained from an event-based SLAM system.

We conduct an additional experiment where the original DERD-Net depth predictions were used to re-estimate the camera poses (in an offline manner, using the camera tracking module in [14]). The resulting poses were then used to build DSIs on which DERD-Net was evaluated. This "reprojection" loop allows us to assess, using standard depth-based metrics, the robustness of our method to noise in camera poses introduced by DERD-Net's own depth inaccuracies. The results are reported in the bottom rows of Tab. 5. The performance shows only minor degradation, particularly for $F_{orig}$, with no metric worsening by more than 13%. Remarkably, even under such self-induced pose noise,

Table 5: Depth estimation performance on *DSEC zurich_city_04_a* using poses computed by ES-PTAM or by camera tracking on DERD-Net's output ("Reprojection" rows). Relative changes with respect to Tab. 3, which reports results obtained using GT poses, are presented in parentheses.

| Algorithm | Poses | Filter | Mean Err [m] $\downarrow$ | Median Err [m] $\downarrow$ | bad-pix [%] $\downarrow$ | #Points [million] $\uparrow$ |
|---|---|---|---|---|---|---|
| DERD-Net | ES-PTAM | $F_{\text{orig}}$ | 1.56 (-3.11%) | 0.45 (-2.17%) | 3.84 (-6.8%) | 1.61 (-3.59%) |
| DERD-Net | ES-PTAM | $F_{\text{denser}}$ | 1.74 (-3.33%) | 0.52 (-3.7%) | 4.84 (-3.97%) | 4.49 (-3.23%) |
| DERD-Net | Reprojection | $F_{\text{orig}}$ | 1.66 (+3.11%) | 0.49 (+6.52%) | 4.19 (+1.7%) | 1.46 (-12.57%) |
| DERD-Net | Reprojection | $F_{\text{denser}}$ | 1.95 (+8.33%) | 0.60 (+11.11%) | 5.65 (+12.1%) | 4.09 (-11.85%) |

DERD-Net's depth estimation errors remain roughly 50% lower than those of prior SOTA methods using ideal poses. These results demonstrate the practical viability of deploying DERD-Net as a depth-estimation module within a self-sustaining SLAM system. Overall, our experiments confirm that DERD-Net remains remarkably robust even when the input poses are significantly degraded, as would be expected in real-world scenarios. See also Appendix Secs. A.3 and A.4

### 4.7 Runtime

Our network achieved an average inference time of only 0.37 ms per Sub-DSI on an NVIDIA RTX A6000. Since predictions are made independently per pixel, inference for each Sub-DSI can be parallelized on the GPU. The total inference time to estimate a depth map of average density (500 pixels) from MVSEC with $F_{\text{orig}}$ is 1.12 ms.

Taking MVSEC as an example (DAVIS cameras of $346 \times 260$ pixels) and DSIs back-projecting 2 million events onto $D = 100$ depth planes, then each DSI creation takes $\approx$45 ms, DSI fusion takes $\approx$26 ms, and pixel selection takes $\approx$0.2 ms on an 8-core computer with Intel Xeon(R) W-2225 CPU operating at 4.10 GHz. These values are common for both the state-of-the-art method MC-EMVS and DERD-Net. It has been shown that DSI creation does not hamper real-time performance [53] because the 3D map can be updated infrequently and on-demand.

Our network adds only a very small runtime compared to the DSI creation time. This ultra-fast performance, combined with its lightweight architecture, enables efficient execution, making DERD-Net ideal for real-world applications requiring low-latency depth estimation.

## 5 Conclusion

We have developed the first learning-based multi-view stereo method for event-based depth estimation. Our approach combines input camera poses with events to produce intermediate geometric representations (DSIs) from which depth is estimated using deep learning. It is directly applicable to both monocular and stereo camera setups. By processing small independent subregions of DSIs in parallel, the framework operates independently of camera resolution and facilitates an efficient network under 1 MB in size with an inference time of only 0.37 ms.

Our framework consistently demonstrated superior performance across several metrics compared to other stereo methods and achieved comparable performance when using purely monocular data. It is the first learning-based depth estimation approach that reports robust generalization on all three *indoor flying* sequences of the MVSEC dataset. Adaptability to different scenes was confirmed on the outdoor driving DSEC dataset, for which it drastically outperformed benchmark approaches across all metrics. Moreover, our framework significantly increased the number of points for which depth can be robustly estimated from DSIs. It also showed strong robustness to noise in camera poses.

Given its exceptional performance, ultra-lightweight architecture, scalability and flexibility across different configurations, our method holds strong potential to become a standard approach for learning depth from events and is highly suitable for real-world robotic applications requiring low latency and low memory such as SLAM [15].

## Acknowledgments

Funded by the Deutsche Forschungsgemeinschaft (DFG, German Research Foundation) under Germany's Excellence Strategy – EXC 2002/1 "Science of Intelligence" – project number 390523135.

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

# A  Appendix: Sensitivity Analyses

In this section we report additional experiments to assess our method's robustness to changes in hyperparameter (Tab. 6), changes in the scene (Tab. 7) and noise in camera poses (Tabs. 8 and 9). We also complement the evaluation on the downstream task of camera tracking (Tab. 10).

## A.1  Sensitivity with respect to Sub-DSI Size

The hyperparameters used for our experiments are detailed in Tab. 2. The LiDAR's sampling interval, the estimated minimum and maximum depths, and the number of depth layers $D$ have all been defined to match the protocol in [12]. In Sec. 4.3, we already provided a comparison between the two AGT filters $F_{\text{orig}}$ and $F_{\text{denser}}$. In this section, we therefore analyze the impact of the size of the Sub-DSI.

We retrained the network for one epoch on the *indoor_flying 2* and *3* sequences and compared its test performance on sequence 1 using radii $r_W = r_H$ of 2, 3, and 4 px, effectively creating Sub-DSI frames of size $5 \times 5$, $7 \times 7$, and $9 \times 9$ px, respectively. Layer dimensions were adapted, while the overall network architecture remained fixed.

Table 6:  Sensitivity analysis of DERD-Net's performance for different Sub-DSI frame sizes after one epoch of training. *MVSEC indoor_flying_1*.

| Sub-DSI frame size | Modality | Mean Err [cm]↓ | Median Err [cm]↓ | bad-pix [%]↓ | SILog Err ×100↓ | AErrR [%]↓ | log RMSE ×100↓ | $\delta < 1.25$ [%]↑ | $\delta < 1.25^2$ [%]↑ | $\delta < 1.25^3$ [%]↑ | #Points [million]↑ |
|---|---|---|---|---|---|---|---|---|---|---|---|
| $5 \times 5$ | stereo + $F_{\text{orig}}$ | 13.92 | 6.51 | 0.96 | 1.20 | 5.71 | 10.97 | 96.07 | 98.56 | 99.60 | 0.99 |
| $7 \times 7$ | stereo + $F_{\text{orig}}$ | 12.13 | 5.82 | 0.85 | 0.97 | 5.04 | 9.93 | 96.81 | 98.85 | 99.66 | 0.98 |
| $9 \times 9$ | stereo + $F_{\text{orig}}$ | 11.58 | 5.54 | 0.82 | 0.92 | 4.85 | 9.64 | 96.97 | 98.91 | 99.68 | 0.98 |
| $5 \times 5$ | stereo + $F_{\text{denser}}$ | 18.42 | 7.88 | 1.79 | 2.10 | 7.64 | 14.50 | 93.71 | 97.58 | 99.10 | 3.03 |
| $7 \times 7$ | stereo + $F_{\text{denser}}$ | 15.41 | 6.80 | 1.35 | 1.55 | 6.34 | 12.56 | 95.24 | 98.20 | 99.36 | 3.01 |
| $9 \times 9$ | stereo + $F_{\text{denser}}$ | 14.59 | 6.39 | 1.30 | 1.48 | 6.07 | 12.21 | 95.52 | 98.29 | 99.37 | 2.99 |

From the results reported in Tab. 6 it can be inferred that performance appears to improve as the radii increase. The network was able to achieve better results after a single epoch using a frame size of $9 \times 9$ than when fully trained on $7 \times 7$ frames (see Tab. 12 in this Appendix), highlighting its potential for further performance improvements.

Nevertheless, increasing the frame size to $9 \times 9$ yielded a reduction of 5% in MAE for both filters after a single training epoch. In contrast to that, the network had to apply 65% more 3D-convolutional operations and its total amount of parameters raised from 70k to 270k. We therefore decided for a $7 \times 7$ frame size for this study. Future research could explore the evident potential to further boost performance by optimizing the sub-DSI size, considering the trade-off between accuracy, parameter count and computational costs.

## A.2  Sensitivity with respect to DSI Transformations

Next, we analyze the robustness of DERD-Net with respect to transformations of the DSI, in particular axis-aligned reflections of the DSIs generated from the *indoor_flying_1* sequence of the MVSEC dataset. Specifically, we flipped the DSIs horizontally, vertically, and both horizontally and vertically. These transformations effectively generate scenes with similar geometric properties (e.g., distance ranges) but novel spatial configurations. This allows us to evaluate how well the network generalizes to previously unseen, yet structurally similar environments. We therefore purposely used *no data augmentation during training* to ensure a representative assessment of the network's inherent robustness. Analogous to previous experiments, we used the single-pixel network that was trained solely on the original *indoor_flying 2* and *3* for evaluation. No retraining was performed.

The results of these experiments are displayed in Tab. 7. Performance worsened only slightly, with results that still significantly outperform all SOTA methods for all tested configurations, indicating that our network might effectively generalize to scenes that share similar depth ranges and texture with those on which it was originally trained.

Table 7: Performance of DERD-Net when applying axis-aligned reflections. *MVSEC indoor_flying_1*.

| Reflection | Modality | Mean Err [cm] ↓ | Median Err [cm] ↓ | bad-pix [%] ↓ | SILog Err ×100 ↓ | AErrR [%] ↓ | log RMSE ×100 ↓ | $\delta < 1.25$ [%] ↑ | $\delta < 1.25^2$ [%] ↑ | $\delta < 1.25^3$ [%] ↑ | #Points [million] ↑ |
|---|---|---|---|---|---|---|---|---|---|---|---|
| none | stereo + $F_{\mathrm{orig}}$ | 11.69 | 5.42 | 0.86 | 0.97 | 4.90 | 9.88 | 96.87 | 98.85 | 99.65 | 0.98 |
| vertical | stereo + $F_{\mathrm{orig}}$ | 12.81 | 6.27 | 0.92 | 1.06 | 5.43 | 10.34 | 96.56 | 98.80 | 99.64 | 0.98 |
| horizontal | stereo + $F_{\mathrm{orig}}$ | 14.26 | 6.81 | 1.01 | 1.23 | 5.88 | 11.08 | 95.85 | 98.56 | 99.60 | 0.98 |
| horizontal + vertical | stereo + $F_{\mathrm{orig}}$ | 14.28 | 6.79 | 1.02 | 1.23 | 5.92 | 11.11 | 95.79 | 98.53 | 99.60 | 0.98 |
| none | stereo + $F_{\mathrm{denser}}$ | 14.86 | 6.47 | 1.31 | 1.49 | 6.14 | 12.30 | 95.45 | 98.24 | 99.37 | 3.01 |
| vertical | stereo + $F_{\mathrm{denser}}$ | 16.27 | 7.42 | 1.44 | 1.65 | 6.79 | 12.96 | 94.91 | 98.09 | 99.31 | 3.01 |
| horizontal | stereo + $F_{\mathrm{denser}}$ | 18.23 | 7.87 | 1.60 | 1.91 | 7.32 | 13.84 | 93.73 | 97.71 | 99.25 | 3.01 |
| horizontal + vertical | stereo + $F_{\mathrm{denser}}$ | 18.86 | 8.37 | 1.68 | 1.98 | 7.64 | 14.14 | 93.46 | 97.61 | 99.22 | 3.01 |

## A.3 Sensitivity with respect to Noise in Camera Poses

In Section Sec. 4.6 we summarized the sensitivity of DERD-Net with respect to noise in the camera poses used to build DSIs, on DSEC data. For completeness, we now show results on MVSEC data.

We repeat the same experiment as that in the top rows of Tab. 5 on the *MVSEC indoor_flying_1* sequence, for which ES-PTAM reported an ATE of 14.93 cm over a 6 m depth range. The results shown in Tab. 8, and compared to those obtained with GT poses in Tab. 12, again highlight DERD-Net's strong robustness to noisy poses estimated by an event-based SLAM system: the MAE and MedAE increased only slightly, while the bad-pix metric even improved. The most pronounced decline was in the number of evaluated points. Nevertheless, DERD-Net with $F_{\mathrm{denser}}$ still predicts depth for 69% more pixels than MC-EMVS with $F_{\mathrm{orig}}$ under ideal poses, while achieving a 30% lower MAE. For its multi-pixel variant, DERD-Net evaluated with noisy poses maintains superior performance over all state-of-the-art methods using GT poses, while still predicting the highest number of points.

Table 8: Quantitative depth estimation performance on *MVSEC indoor_flying_1* using poses computed downstream of ES-PTAM. Relative changes with respect to Tab. 12, which reports results obtained using GT poses, are presented in parentheses.

| Algorithm | Poses | Filter | Mean Err [m] ↓ | Median Err [m] ↓ | bad-pix [%] ↓ | #Points [million] ↑ |
|---|---|---|---|---|---|---|
| DERD-Net | ES-PTAM | $F_{\mathrm{orig}}$ | 12.72 (+8.81%) | 6.33 (+16.79%) | 0.65 (-24.42%) | 0.68 (-30.61%) |
| DERD-Net | ES-PTAM | $F_{\mathrm{denser}}$ | 15.76 (+6.06%) | 7.36 (+13.76%) | 1.17 (-10.69%) | 1.62 (-46.18%) |
| DERD-Net (multi-pixel) | ES-PTAM | $F_{\mathrm{orig}}$ | 13.53 (+10.27%) | 6.76 (+19.01%) | 0.65 (-24.42%) | 3.41 (-33.91%) |
| DERD-Net (multi-pixel) | ES-PTAM | $F_{\mathrm{denser}}$ | 16.60 (+6.62%) | 7.65 (+16.08%) | 1.21 (-11.68%) | 6.27 (-46.46%) |

In the interest of thoroughness, we also used poses from the state-of-the-art event-based stereo visual–inertial odometry system ESVO2 [42], which is notably more accurate than ES-PTAM, to run DERD-Net on *MVSEC indoor_flying_1*, *indoor_flying_2*, and *indoor_flying_3*. The mean results are reported in Tab. 9. Compared to Tab. 3, DERD-Net shows only a slight decrease in depth estimation performance on $F_{\mathrm{orig}}$, while it even improves on $F_{\mathrm{denser}}$, confirming its robustness to pose noise from an event-based SLAM system integrating events and inertial data.

Table 9: Quantitative depth estimation performance averaged over *MVSEC indoor_flying_1, _2*, and *_3* using poses computed downstream of ESVO2. Relative changes with respect to Tab. 3, which reports results obtained using GT poses, are presented in parentheses.

| Algorithm | Poses | Filter | Mean Err [m] ↓ | Median Err [m] ↓ | bad-pix [%] ↓ | #Points [million] ↑ |
|---|---|---|---|---|---|---|
| DERD-Net | ESVO2 | $F_{\mathrm{orig}}$ | 11.53 (-1.37%) | 5.73 (+4.18%) | 1.09 (+22.47%) | 0.61 (-22.78%) |
| DERD-Net | ESVO2 | $F_{\mathrm{denser}}$ | 13.69 (-10.17%) | 6.32 (-5.39%) | 1.46 (-14.12%) | 1.45 (-47.65%) |

### A.4 Downstream Camera Tracking Performance Analysis

As intermediate results to those in the bottom part of Tab. 5, we report offline camera tracking performance using the edge-alignment camera tracking module in [14] acting on input events and the local maps built using DERD-Net's depth predictions (from GT poses). Camera tracking performance is given in terms of ATE and Absolute Rotation Error (ARE) on the DSEC [13] driving dataset in Tab. 10. Although our model was trained only on the *Zurich_City_04_a* sequence, we evaluate it on all *Zurich_City_04* sequences to highlight its generalization capabilities.

Table 10: Camera tracking performance on *DSEC zurich_city_04, without DERD-Net retraining.*

| Sequence | zc04a | zc04b | zc04c | zc04d | zc04e | zc04f |
|---|---|---|---|---|---|---|
| Duration [s] | 35 | 13.4 | 53 | 47.8 | 13.6 | 43.1 |
| ATE RMSE [cm] $\downarrow$ | 17.07 | 7.85 | 14.00 | 55.64 | 5.71 | 36.11 |
| ARE RMSE [deg] $\downarrow$ | 0.31 | 0.08 | 0.45 | 0.67 | 0.11 | 0.72 |

The obtained pose errors in the 50 m depth range scenes across all sequences show strong performance of DERD-Net for downstream tasks such as pose estimation via simple photometric edge alignment on event images, as well as its robust generalization even when trained on a single sequence. Training on a more diverse set of DSEC sequences would be expected to further enhance these results. These values are not comparable to online SLAM tracking results because they assume that the 3D map was pre-built offline using DERD-Net with GT poses. Therefore, the estimated camera poses reported here do not accumulate drift.

## B Appendix - Detailed per-Sequence Results

The average results of different SOTA methods compared to DERD-Net are presented in Tab. 11. In Tabs. 12 to 14, the individual performance on each of the respective sequences *indoor_flying 1, 2, 3* from the MVSEC dataset are displayed. Table 15 presents the corresponding results for the *Zurich_City_04_a* sequence from the DSEC dataset.

**MVSEC Averaged**

Table 11: Quantitative comparison of the proposed methods with the state of the art. *MVSEC indoor_flying* (average).

| | Algorithm | Modality | Mean Err [cm] $\downarrow$ | Median Err [cm] $\downarrow$ | bad-pix [%] $\downarrow$ | SILog Err $\times100$ $\downarrow$ | AErrR [%] $\downarrow$ | log RMSE $\times100$ $\downarrow$ | $\delta < 1.25$ [%] $\uparrow$ | $\delta < 1.25^2$ [%] $\uparrow$ | $\delta < 1.25^3$ [%] $\uparrow$ | #Points [million] $\uparrow$ |
|---|---|---|---|---|---|---|---|---|---|---|---|---|
| SOTA | EMVS [11] | monocular + $F_{orig}$ | 33.78 | 14.35 | 3.84 | 4.20 | 12.74 | 20.72 | 84.75 | 94.87 | 97.99 | 1.27 |
| | EMVS [11] | monocular + $F_{denser}$ | 50.32 | 20.81 | 11.46 | 11.37 | 20.87 | 33.75 | 73.43 | 88.09 | 93.71 | 4.15 |
| | ESVO [19] | stereo | 25.00 | 10.59 | 3.35 | 3.48 | 10.19 | 18.83 | 90.44 | 95.76 | 97.98 | 2.04 |
| | ESVO indep. 1s | stereo | 22.70 | 9.83 | 2.83 | 3.03 | 9.59 | 17.53 | 91.82 | 96.50 | 98.38 | 1.56 |
| | SGM indep. 1s | stereo | 35.42 | 12.35 | 6.39 | 8.45 | 16.17 | 29.49 | 85.34 | 93.05 | 96.03 | **14.46** |
| | GTS indep. 1s | stereo | 389.00 | 45.43 | 38.45 | 74.47 | 102.92 | 89.08 | 49.56 | 62.19 | 69.36 | 0.06 |
| | MC-EMVS [12] | stereo + $F_{orig}$ | 20.07 | 9.53 | 1.35 | 1.72 | 7.80 | 13.24 | 95.04 | 98.08 | 99.21 | 0.81 |
| | MC-EMVS [12] | stereo + $F_{denser}$ | 28.38 | 12.38 | 3.26 | 3.43 | 10.94 | 18.60 | 89.41 | 96.09 | 98.33 | 2.77 |
| | MC-EMVS [12] + MF | stereo + $F_{orig}$ | 20.64 | 9.72 | 1.43 | 1.80 | 7.94 | 13.54 | 94.74 | 97.95 | 99.17 | 3.00 |
| Ours | DERD-Net | monocular + $F_{orig}$ | 23.68 | 11.55 | 2.78 | 2.62 | 10.18 | 16.20 | 90.25 | 97.36 | 99.02 | 1.21 |
| | DERD-Net | monocular + $F_{denser}$ | 28.52 | 13.85 | 4.87 | 3.77 | 12.33 | 19.46 | 85.78 | 95.77 | 98.50 | 4.15 |
| | DERD-Net without EL | stereo + $F_{orig}$ | 12.00 | 5.73 | 0.92 | 0.98 | 5.15 | 9.92 | 96.99 | 98.86 | 99.63 | 0.79 |
| | DERD-Net | stereo + $F_{orig}$ | **11.69** | **5.50** | **0.89** | **0.96** | **5.05** | **9.83** | **96.99** | **98.89** | **99.64** | 0.79 |
| | DERD-Net | stereo + $F_{denser}$ | 15.24 | 6.68 | 1.70 | 1.54 | 6.41 | 12.44 | 95.00 | 98.19 | 99.39 | 2.77 |
| | DERD-Net multi-pixel | stereo + $F_{orig}$ | 12.02 | 5.63 | 0.90 | 0.99 | 5.13 | 9.94 | 96.89 | 98.83 | 99.63 | 4.32 |
| | DERD-Net multi-pixel | stereo + $F_{denser}$ | 15.68 | 6.73 | 1.74 | 1.59 | 6.54 | 12.61 | 94.75 | 98.10 | 99.36 | 11.33 |

## MVSEC Indoor Flying 1

Table 12: Quantitative comparison of the proposed methods with the state of the art. *MVSEC indoor_flying_1*.

| Algorithm | Modality | Mean Err [cm]↓ | Median Err [cm]↓ | bad-pix [%]↓ | SILog Err ×100↓ | AErrR [%]↓ | log RMSE ×100↓ | $\delta < 1.25$ [%]↑ | $\delta < 1.25^2$ [%]↑ | $\delta < 1.25^3$ [%]↑ | #Points [million]↑ |
|---|---|---|---|---|---|---|---|---|---|---|---|
| EMVS [11] | monocular + $F_{orig}$ | 39.37 | 14.95 | 3.05 | 4.72 | 13.25 | 22.10 | 82.03 | 93.43 | 97.62 | 1.21 |
| EMVS [11] | monocular + $F_{denser}$ | 60.65 | 24.20 | 12.56 | 13.88 | 24.49 | 37.29 | 69.04 | 84.58 | 91.54 | 4.64 |
| ESVO [19] | stereo | 24.04 | 10.21 | 2.54 | 2.94 | 9.76 | 17.17 | 91.43 | 96.53 | 98.55 | 1.95 |
| ESVO indep. 1s | stereo | 23.39 | 10.03 | 2.18 | 2.79 | 9.78 | 16.72 | 91.57 | 96.84 | 98.79 | 1.41 |
| SGM indep. 1s | stereo | 35.45 | 13.61 | 5.54 | 7.35 | 15.03 | 27.46 | 85.96 | 93.51 | 96.40 | 11.64 |
| GTS indep. 1s | stereo | 700.37 | 38.39 | 32.51 | 79.26 | 111.21 | 91.44 | 54.27 | 67.16 | 73.39 | 0.03 |
| MC-EMVS [12] | stereo + $F_{orig}$ | 22.53 | 9.72 | 1.30 | 1.94 | 7.91 | 14.11 | 93.49 | 97.50 | 99.17 | 0.96 |
| MC-EMVS [12] | stereo + $F_{denser}$ | 31.43 | 13.14 | 3.11 | 3.99 | 11.69 | 20.03 | 88.16 | 95.28 | 97.91 | 3.01 |
| MC-EMVS [12] + MF | stereo + $F_{orig}$ | 23.33 | 9.90 | 1.39 | 2.08 | 8.12 | 14.61 | 93.16 | 97.28 | 99.05 | 3.48 |
| DERD-Net | monocular + $F_{orig}$ | 25.76 | 12.60 | 1.89 | 2.62 | 10.39 | 16.19 | 89.42 | 97.60 | 99.24 | 1.50 |
| DERD-Net | monocular + $F_{denser}$ | 30.90 | 15.00 | 3.34 | 3.86 | 12.62 | 19.65 | 85.37 | 95.94 | 98.61 | 4.64 |
| DERD-Net without EL | stereo + $F_{orig}$ | 12.05 | 5.65 | 0.86 | 0.97 | 4.98 | 9.90 | 96.89 | 98.85 | 99.66 | 0.98 |
| DERD-Net | stereo + $F_{orig}$ | 11.69 | 5.42 | 0.86 | 0.97 | 4.90 | 9.88 | 96.87 | 98.85 | 99.65 | 0.98 |
| DERD-Net | stereo + $F_{denser}$ | 14.86 | 6.47 | 1.31 | 1.49 | 6.14 | 12.30 | 95.45 | 98.24 | 99.37 | 3.01 |
| DERD-Net multi-pixel | stereo + $F_{orig}$ | 12.27 | 5.68 | 0.86 | 0.99 | 5.06 | 9.98 | 96.76 | 98.77 | 99.65 | 5.16 |
| DERD-Net multi-pixel | stereo + $F_{denser}$ | 15.57 | 6.59 | 1.37 | 1.58 | 6.39 | 12.61 | 95.13 | 98.11 | 99.33 | 11.71 |

## MVSEC Indoor Flying 2

Table 13: Quantitative comparison of the proposed methods with the state of the art. *MVSEC indoor_flying_2*.

| Algorithm | Modality | Mean Err [cm]↓ | Median Err [cm]↓ | bad-pix [%]↓ | SILog Err ×100↓ | AErrR [%]↓ | log RMSE ×100↓ | $\delta < 1.25$ [%]↑ | $\delta < 1.25^2$ [%]↑ | $\delta < 1.25^3$ [%]↑ | #Points [million]↑ |
|---|---|---|---|---|---|---|---|---|---|---|---|
| EMVS [11] | monocular + $F_{orig}$ | 31.42 | 13.01 | 6.15 | 4.56 | 13.37 | 21.80 | 84.07 | 94.72 | 97.88 | 1.17 |
| EMVS [11] | monocular + $F_{denser}$ | 45.69 | 17.96 | 14.66 | 11.74 | 19.86 | 34.81 | 72.69 | 88.27 | 94.01 | 3.65 |
| ESVO [19] | stereo | 21.34 | 8.97 | 3.75 | 3.48 | 9.32 | 19.14 | 91.60 | 95.88 | 97.86 | 1.89 |
| ESVO indep. 1s | stereo | 20.42 | 8.63 | 3.50 | 3.24 | 9.14 | 18.35 | 92.03 | 96.19 | 98.19 | 1.41 |
| SGM indep. 1s | stereo | 32.94 | 8.75 | 8.29 | 9.50 | 15.82 | 31.54 | 84.40 | 92.33 | 95.48 | 16.95 |
| GTS indep. 1s | stereo | 167.14 | 37.23 | 43.08 | 71.91 | 94.78 | 86.93 | 49.36 | 60.54 | 67.76 | 0.07 |
| MC-EMVS [12] | stereo + $F_{orig}$ | 18.20 | 8.49 | 1.77 | 1.78 | 8.13 | 13.59 | 95.53 | 98.13 | 99.08 | 0.65 |
| MC-EMVS [12] | stereo + $F_{denser}$ | 25.81 | 10.34 | 4.65 | 3.48 | 10.89 | 18.91 | 89.10 | 95.93 | 98.35 | 2.25 |
| MC-EMVS [12] + MF | stereo + $F_{orig}$ | 18.58 | 8.68 | 1.86 | 1.81 | 8.19 | 13.71 | 95.27 | 98.07 | 99.09 | 2.42 |
| DERD-Net | monocular + $F_{orig}$ | 23.37 | 10.43 | 4.98 | 3.31 | 11.07 | 18.41 | 88.30 | 96.23 | 98.46 | 0.98 |
| DERD-Net | monocular + $F_{denser}$ | 27.65 | 12.75 | 8.68 | 4.60 | 13.39 | 21.76 | 82.75 | 94.34 | 97.90 | 3.65 |
| DERD-Net without EL | stereo + $F_{orig}$ | 11.44 | 5.23 | 1.34 | 1.13 | 5.45 | 10.67 | 96.67 | 98.60 | 99.52 | 0.58 |
| DERD-Net | stereo + $F_{orig}$ | 11.11 | 4.94 | 1.26 | 1.10 | 5.34 | 10.50 | 96.69 | 98.66 | 99.54 | 0.58 |
| DERD-Net | stereo + $F_{denser}$ | 14.46 | 5.92 | 2.78 | 1.72 | 6.74 | 13.17 | 94.05 | 97.88 | 99.32 | 2.25 |
| DERD-Net multi-pixel | stereo + $F_{orig}$ | 11.29 | 4.88 | 1.28 | 1.14 | 5.39 | 10.66 | 96.50 | 98.61 | 99.54 | 3.21 |
| DERD-Net multi-pixel | stereo + $F_{denser}$ | 14.92 | 5.95 | 2.85 | 1.80 | 6.86 | 13.47 | 93.67 | 97.76 | 99.29 | 9.52 |

## MVSEC Indoor Flying 3

Table 14: Quantitative comparison of the proposed methods with the state of the art. *MVSEC indoor_flying_3*.

| Algorithm | Modality | Mean Err [cm]↓ | Median Err [cm]↓ | bad-pix [%]↓ | SILog Err ×100↓ | AErrR [%]↓ | log RMSE ×100↓ | $\delta < 1.25$ [%]↑ | $\delta < 1.25^2$ [%]↑ | $\delta < 1.25^3$ [%]↑ | #Points [million]↑ |
|---|---|---|---|---|---|---|---|---|---|---|---|
| EMVS [11] | monocular + $F_{orig}$ | 30.54 | 15.09 | 2.31 | 3.33 | 11.59 | 18.27 | 88.16 | 96.45 | 98.47 | 1.42 |
| EMVS [11] | monocular + $F_{denser}$ | 44.62 | 20.26 | 7.15 | 8.50 | 18.26 | 29.15 | 78.55 | 91.41 | 95.57 | 4.15 |
| ESVO [19] | stereo | 29.62 | 12.61 | 3.78 | 4.02 | 11.50 | 20.20 | 84.88 | 94.88 | 97.52 | 2.29 |
| ESVO indep. 1s | stereo | 24.29 | 10.84 | 2.81 | 3.05 | 9.84 | 17.54 | 91.87 | 96.46 | 98.16 | 1.86 |
| SGM indep. 1s | stereo | 37.86 | 14.69 | 5.33 | 8.52 | 17.65 | 29.46 | 85.67 | 93.31 | 96.21 | 14.81 |
| GTS indep. 1s | stereo | 299.48 | 60.66 | 39.75 | 72.24 | 102.77 | 88.87 | 45.04 | 58.86 | 66.94 | 0.08 |
| MC-EMVS [12] | stereo + $F_{orig}$ | 19.49 | 10.38 | 0.99 | 1.43 | 7.35 | 12.01 | 96.09 | 98.60 | 99.38 | 0.82 |
| MC-EMVS [12] | stereo + $F_{denser}$ | 27.89 | 13.65 | 2.01 | 2.83 | 10.25 | 16.85 | 90.97 | 97.05 | 98.73 | 3.04 |
| MC-EMVS [12] + MF | stereo + $F_{orig}$ | 20.02 | 10.59 | 1.02 | 1.50 | 7.50 | 12.30 | 95.79 | 98.51 | 99.36 | 3.11 |
| DERD-Net | monocular + $F_{orig}$ | 21.91 | 11.62 | 1.46 | 1.93 | 9.07 | 14.01 | 93.02 | 98.25 | 99.36 | 1.14 |
| DERD-Net | monocular + $F_{denser}$ | 27.01 | 13.80 | 2.58 | 2.85 | 10.99 | 16.96 | 89.23 | 97.04 | 98.98 | 4.15 |
| DERD-Net without EL | stereo + $F_{orig}$ | 12.50 | 6.31 | 0.57 | 0.84 | 5.03 | 9.20 | 97.41 | 99.13 | 99.72 | 0.82 |
| DERD-Net | stereo + $F_{orig}$ | 12.28 | 6.13 | 0.55 | 0.82 | 4.91 | 9.11 | 97.41 | 99.15 | 99.74 | 0.82 |
| DERD-Net | stereo + $F_{denser}$ | 16.39 | 7.64 | 1.02 | 1.40 | 6.36 | 11.84 | 95.49 | 98.45 | 99.48 | 3.04 |
| DERD-Net multi-pixel | stereo + $F_{orig}$ | 12.50 | 6.34 | 0.56 | 0.84 | 4.93 | 9.17 | 97.41 | 99.12 | 99.71 | 4.59 |
| DERD-Net multi-pixel | stereo + $F_{denser}$ | 16.55 | 7.65 | 1.01 | 1.38 | 6.36 | 11.76 | 95.44 | 98.43 | 99.47 | 12.77 |

**DSEC**

Table 15: Quantitative comparison of the proposed methods with the state of the art. *DSEC Zurich_City_04_a*.

| | Algorithm | Modality | Mean Err [m]↓ | Median Err [m]↓ | bad-pix [%]↓ | SILog Err ×100↓ | AErrR [%]↓ | log RMSE ×100↓ | $\delta < 1.25$ [%]↑ | $\delta < 1.25^2$ [%]↑ | $\delta < 1.25^3$ [%]↑ | #Points [million]↑ |
|---|---|---|---|---|---|---|---|---|---|---|---|---|
| SOTA | EMVS [11] | monocular + $F_{orig}$ | 5.64 | 2.52 | 13.68 | 13.23 | 25.52 | 36.49 | 72.56 | 87.12 | 93.56 | 1.31 |
| | EMVS [11] | monocular + $F_{denser}$ | 7.01 | 3.56 | 24.33 | 23.07 | 41.74 | 48.52 | 63.00 | 79.81 | 87.71 | 6.09 |
| | ESVO [19] | stereo | 3.93 | 1.62 | 10.54 | 8.30 | 17.66 | 28.90 | 84.37 | 92.81 | 96.05 | 9.40 |
| | SGM [38] | stereo | 6.74 | 1.58 | 15.25 | 17.95 | 18.42 | 42.51 | 80.66 | 89.12 | 93.16 | 8.30 |
| | GTS [39] | stereo | 26.24 | 1.62 | 32.56 | 61.58 | 33.45 | 79.26 | 68.07 | 78.39 | 85.85 | 0.11 |
| | MC-EMVS [12] | stereo + $F_{orig}$ | 3.27 | 0.90 | 10.75 | 8.19 | 17.48 | 28.73 | 83.30 | 91.56 | 95.62 | 1.25 |
| | MC-EMVS [12] | stereo + $F_{denser}$ | 4.76 | 1.56 | 17.42 | 15.84 | 30.67 | 40.45 | 76.37 | 86.01 | 90.97 | 4.64 |
| | MC-EMVS [12] + MF | stereo + $F_{orig}$ | 3.51 | 0.96 | 11.81 | 8.89 | 18.84 | 29.99 | 81.72 | 90.68 | 95.07 | 3.83 |
| Ours | DERD-Net | monocular + $F_{orig}$ | 3.12 | 1.60 | 5.50 | 3.96 | 12.19 | 19.92 | 86.06 | 96.29 | 98.61 | 2.10 |
| | DERD-Net | monocular + $F_{denser}$ | 3.01 | 1.50 | 6.35 | 4.04 | 12.24 | 20.12 | 86.46 | 96.07 | 98.41 | 6.09 |
| | DERD-Net | stereo + $F_{orig}$ | 1.61 | **0.46** | 4.12 | 2.78 | 7.03 | 16.68 | 93.50 | 97.05 | 98.66 | 1.67 |
| | DERD-Net | stereo + $F_{denser}$ | 1.80 | 0.54 | 5.04 | 2.91 | 7.59 | 17.06 | 92.09 | 96.72 | 98.56 | 4.64 |
| | DERD-Net multi-pixel | stereo + $F_{orig}$ | **1.59** | 0.47 | **3.81** | **2.54** | **6.76** | **15.93** | **93.60** | **97.18** | **98.78** | 6.59 |
| | DERD-Net multi-pixel | stereo + $F_{denser}$ | 1.79 | 0.54 | 4.61 | 2.76 | 7.46 | 16.62 | 92.31 | 96.82 | 98.62 | **14.74** |

