# OpenReview forum: "DERD-Net: Learning Depth from Event-based Ray Densities"
_NeurIPS.cc/2025/Conference — NeurIPS 2025 spotlight_

### Official Review · Reviewer_gXBu · 2025-06-30

**Clarity:** 3
**Significance:** 4
**Originality:** 3
**Rating:** 5
**Confidence:** 4

**Summary:**

The paper addresses the problem of depth estimation (both monocular and stereo) from event-camera images. The proposed method is pertinent because it uses an important modality (event-camera outputs) to estimate depth, which is fundamental to computer vision. It uses a patch-like formulation, efficient learning based processing, and a model that is a combination of 3D Convolutions and GRUs to achieve its desired objectives.  The paper reports very convincing metrics on multiple datasets making it a significant improvement over the prior art.

**Questions:**

+ What is the rationale behind the patch-like formulation and why can't vanilla 3D-Convolution achieve the same effect?
+ How sensitive is the model's performance to errors in camera pose estimation?
+ Can this method be potentially employed for real-time SLAM?

**Ethical Concerns:**

["NO or VERY MINOR ethics concerns only"]

**Final Justification:**

I was already fairly convinced about the quality of this paper and the rebuttal has convinced me further in this regard.

**Limitations:**

It would have been better if the authors had consolidated all limitations at one place.

**Paper Formatting Concerns:**

None.

**Quality:**

3

**Strengths And Weaknesses:**

Strengths:

+ The method uses a event-cameras, which is an important input modality for depth estimation. The overall pipeline seems  fairly novel and technically sound.
+ The method estimates depth in both monocular and stereo settings.
+ The paper gains considerably in terms of efficiency over the SoTA as shown by its ultra-fast inference time.
+ The patch-like (sub-DSI) formulation seems very pragmatic.
+  The paper presents very well-designed experiments and is overall well-written.

Weaknesses:

+ Talking about the robustness of this method to noisy event-camera measurements and thereby incorporating statistical significance of the metrics could benefit the overall paper quality.

---

> ### Author Rebuttal · Authors · 2025-07-30
>
> Thanks for the positive feedback, especially regarding our method's novelty, efficiency and well-designed experiments. Our work aims to propose learning pipelines targeting the challenges of processing sparse, asynchronous event data for efficient depth estimation.
>
> ## **Robustness to noise in event camera and pose measurements**
> All the experiments in the paper were conducted on real data with real noisy measurements. There is significant noise in the MVSEC flying sequences because it uses an older (mDAVIS346) sensor, and contains interference from the motion capture infrared lights. Thus, our method already shows robustness to realistic noise in the event sensor. Moreover, we evaluate on multiple datasets that use different sensors (DSEC uses Prophesse Gen3 cameras), and obtain SOTA performance under diverse noise characteristics. Although a more systematic noise analysis by injecting artificial random noise could be useful to gauge statistical significance, it is non-standard in 3D reconstruction and thus beyond the scope of the paper.
>
> For robustness against noisy poses, we report results by running DERD-Net on poses estimated by other stereo event-based SLAM pipelines (ES-PTAM and ESVO2) which contain realistic noise and drift.
>
> Our experimental pipeline: Stereo events --> estimated poses using event SLAM --> DSI --> Depth maps using DERD-Net
>
> (a) On DSEC zurich_city_04_a sequence, instead of GT poses (from LiDAR-IMU odometry), we used camera poses estimated by a stereo event odometry system ES-PTAM [1], which reported 131.62 cm Absolute Trajectory Error (ATE) in 50 m depth range scene, to run DERD-Net and achieved the following depth estimation performance:
>
> | Algorithm | Mean Err [m] ↓ | Median Err [m] ↓ | bad-pix [%] ↓ | # Points [million] ↑ |
> |---|---|---|---|---|
> | Noisy DERD-Net + F_orig        | 1.56 (-3.11%) | 0.45 (-2.17%)  | 3.84 (-6.8%)   | 1.61 (-3.59%) |
> | Noisy DERD-Net + F_denser   | 1.74 (-3.33%) | 0.52 (-3.7%)  | 4.84 (-3.97%)   | 4.49 (-3.23%) |
>
> Compared to Table 3, while the number of points for which depth is evaluated slightly decreased, DERD-Net's depth estimation errors actually even improved. This shows strong robustness to noisy poses obtained from an event SLAM system.
>
> (b) On MVSEC flying1, instead of GT poses (from motion capture), we used camera poses estimated by ES-PTAM [1] (ATE 14.93 cm in 6 m depth range scene) to run DERD-Net and achieved the following depth estimation performance:
>
> | Algorithm | Mean Err [cm] ↓ | Median Err [cm] ↓ | bad-pix [%] ↓ | # Points [million] ↑ |
> |---|---|---|---|---|
> | Noisy DERD-Net + F_orig        | 12.72 (+8.81%) | 6.33 (+16.79%)  | 0.65 (-24.42%)   | 0.68 (-30.61%) |
> | Noisy DERD-Net + F_denser   | 15.76 (+6.06%) | 7.36 (+13.76%)  | 1.17 (-10.69%)   | 1.62 (-46.18%) |
> | Noisy Multi-Pixel DERD-Net + F_orig   | 13.53 (+10.27%) |  6.76 (+19.01%) | 0.65 (-24.42%) | 3.41(-33.91%) |
> | Noisy Multi-Pixel DERD-Net + F_denser   | 16.60 (+6.62%) | 7.65 (+16.08%)  |  1.21 (-11.68%) | 6.27 (-46.46%) |
>
> Compared to Table 8, the MAE and MedAE increased only slightly, while bad-pix even improved. The strongest decrease here was in the number of points. However, that Noisy DERD-Net on F_denser still predicts on 69% more pixels than MC-EMVS with f_orig using *ideal* poses, while still yielding a 30% decrease in MAE. For its multi-pixel version, Noisy DERD-Net shows superior performance than all SOTA methods while still predicting on most points. This shows strong robustness to noisy poses obtained from an event SLAM system.
>
> (c)  For completeness, we also used poses from SOTA event-based stereo visual-inertial odometry system ESVO2 [2] to run DERD-Net on MVSEC flying1, flying2 and flying3 and report mean metrics. Compared to ES-PTAM, ESVO2 is notably more accurate due to the visual-inertial data fusion, producing less noisy poses, which yield an average ATE of ~8.3 cm in 6 m depth range scene.
>
> | Algorithm | Mean Err [cm] ↓ | Median Err [cm] ↓ | bad-pix [%] ↓ | # Points [million] ↑ |
> |---|---|---|---|---|
> | Noisy DERD-Net + F_orig        | 11.53 (-1.37%) | 5.73 (+4.18%)  | 1.09 (+22.47%)   | 0.61 (-22.78%) |
> | Noisy DERD-Net + F_denser   | 13.69 (-10.17%) | 6.32 (-5.39%)  | 1.46 (-14.12%)   | 1.45 (-47.65%) |
>
> Compared to Table 3, this shows a slight decrease in performance for DERD-Net's depth estimation error on f_orig, while it even improves on f_denser. Again, the strongest decrease is in the number of predicted points, as observed in (b). This shows robustness to noisy poses obtained from a SLAM system using events and IMU.
>
> ## **Real-time SLAM**
> Our method runs real-time with DAVIS346 cameras.
> On a PC with Intel Xeon(R) W-2225 CPU operating at 4.10GHz with 8 cores, for MVSEC (DAVIS cameras with 346x260 pixels, 100 depth planes, 2 million events), each DSI construction takes ~45 ms, DSI fusion (for stereo) takes ~26 ms, and pixel selection takes ~0.2 ms.
> On top of the above steps, DERD-Net inference for a single sub-DSI takes 0.37 ms. By parallelly processing sub-DSIs in GPU, this amounts to a total time of 1.12 ms for DERD-Net inference.
>
> It has been shown that the stereo SLAM method ES-PTAM [1] which also uses DSI fusion for map can run real-time with DAVIS cameras [3] because the 3D map can be updated infrequently and on-demand. Compared to ES-PTAM, our method needs an additional ~1 ms in total runtime, and thus can also run real-time.
>
> # Answers to Questions:
>
> 1. Processing sub-DSIs / patches instead of full 3D convolutions is preferable because:
>     - We can leverage sparsity of DSI by only processing a subset of the voxels, producing overall faster inference and smaller networks (lower memory, lower training time). This will otherwise be prohibitively expensive for real-time processing.
>     - A purely convolutional network would struggle to model the sequential structure of ray accumulations along the depth dimension of the DSI. This sequential aspect of how rays build up evidence for surfaces along their trajectory is critical for distinguishing true ray intersections (indicative of surfaces) from noise or spurious alignments. Our use of a GRU allows the network to capture this progression explicitly.
>    - A 3D convolution on the full DSI would lead to reduction of training data because each DSI will be a single training sample. Moreover, the network may learn to over-fit to a particular scenario because it considers a large spatial window and derives semantic context, instead of learning about local ray intersection patterns, to estimate depth. Our approach (breaking into smaller sub-DSIs) deals with limited labeled training data (which is a prevalent issue in event vision) by turning each pixel into a training sample, therefore augmenting the available dataset. For instance, each MVSEC sequence only has ~1.5k frames, which would result in only 3k data points (including cross-validation). A bigger vanilla network would make learning from less data more difficult. By creating sub-DSIs, we produce >1 million training samples per training set with only 70k parameters to train.
>    - It is independent of input camera resolution.
>
> 2. Please refer to noise sensitivity analysis above. For further experiments showing robustness, please also see our response to Reviewer PC8y.
>
> 3. Yes, our method can be used for real-time SLAM. Please refer to the section on Real-time SLAM above for runtime. For further details and comparison to SOTA method MC-EMVS, please refer to our Response to Reviewer Yns5.
>
> References:
>
> [1] Ghosh et al. ES-PTAM: Event-Based Stereo Parallel Tracking and Mapping. In ECCVW 2024.
>
> [2] Niu et al. ESVO2: Direct Visual-Inertial Odometry With Stereo Event Cameras. IEEE T-RO 41 (2025), 2164–2183.
>
> [3] Ghosh, Cavinato, Gallego. SLAM with Stereo Event Cameras. ECCV 2024 demo.

---

> ### Comment · Reviewer_gXBu · 2025-08-03
> **Response to Rebuttal**
>
> Dear authors,
>
> Thanks very much for a detailed rebuttal. I am happy to note that all my concerns have been answered. I will maintain my rating.
> I would urge the authors to include all the additional analyses in their final version as well to improve the overall quality.

---

> > ### Author Response · Authors · 2025-08-05
> > **Response to Comment**
> >
> > Thank you very much for your encouraging comment. We’re pleased to hear that all your concerns have been addressed. We will certainly incorporate the additional analyses and results from the rebuttal into the final version of the paper. Thank you again for your feedback that helped improve the quality of our work. Please don’t hesitate to reach out if any additional questions arise.

---

### Official Review · Reviewer_KbkG · 2025-07-02

**Clarity:** 2
**Significance:** 4
**Originality:** 3
**Rating:** 5
**Confidence:** 4

**Summary:**

This paper presents DERD-Net, a neural network for depth estimation from event data captured on a stereo setup. The core idea is to first convert the event stream into a Disparity Space Image (DSI), which captures the spatial density of back-projected event rays (assuming known camera poses).
The two DSIs from the stereo pairs are then fused together with the harmonic mean.
The method then focuses on small, local subregions (Sub-DSIs) around selected pixels, where a lightweight neural network applies 3D convolutions followed by a gated recurrent unit to predict depth from these Sub-DSIs.
Experiments are done on the MVSEC and DSEC datasets, showing that the method DERD-Net outperforms all state-of-the-art baselines. Remarkable, even only monocular input (one DSI), the method could still achieve reasonable results.

**Questions:**

- Could you clarify how depth can be inferred under the monocular setting?
- How sensitive is DERD-Net to noise or drift in camera poses?
- How is the runtime compared to other state-of-the-art methods?
- Have you explored the performance trade-off with larger network architectures?
- Is the performance already saturated with this small model? Or is it possible to improve accuracy by paying more computational costs?

**Ethical Concerns:**

["NO or VERY MINOR ethics concerns only"]

**Final Justification:**

The clarification and additional details are satisfactory. The rating is updated to 5.

**Limitations:**

Yes

**Quality:**

3

**Strengths And Weaknesses:**

Strengths:
- The method outperforms existing methods on standard benchmarks (MVSEC and DSEC).
- The method is low-latency and ideal for real-world applications.
- Surprisingly good depth estimates even from a single event camera.

Weaknesses:
- The paper could be clearer about how the monocular setting works, and how different are the operations from the default stereo setting. The presentation clarity is not very good. More explanation or intuition about why the method still works given monocular data is needed.
- The method's performance is fundamentally dependent on having accurate camera poses to construct the DSI. The paper acknowledges this by stating that poses can be obtained from tracking methods or dead-reckoning. However, it does not analyze the model's sensitivity to pose errors.
- The paper also does not seem to provide runtime comparisons with other state-of-the-art methods.

---

> ### Author Rebuttal · Authors · 2025-07-30
>
> Thanks for your positive feedback and for highlighting that our approach outperforms all SOTA methods on standard benchmarks, and that it even achieves comparable performance when using monocular input only. We appreciate our method being recognized “ideal for real-world applications” due to its low latency. We address the concerns below.
>
> ## **Monocular setting**
>
> Thank you for raising this point. We clarify that the DERD-Net architecture and inference process remain exactly the same in both the monocular and stereo settings. In both cases, DERD-Net operates on local subregions of the respective DSI. The difference lies in how this DSI is constructed. In the monocular setting, the DSI is populated by back-projecting rays from an event camera as described in the first steps of the monocular method EMVS [9] (IJCV 2018). Such a monocular DSI contains depth information due to motion parallax from multiple viewpoints. In the stereo setting, MC-EMVS [10] (AISY 2022), two such DSIs are simply fused into a new DSI using element-wise operations (like Harmonic Mean).
> Figure 4 in MC-EMVS illustrates the differences between monocular and stereo DSIs. The stereo setup makes depth convergence faster. While argmax (in MC-EMVS) relies on a single axis of accumulation and needs stereo cameras for fast depth convergence and reliable depth estimation, DERD-Net leverages deep learning to robustly infer local 3D ray density patterns from DSIs in monocular settings, enabling effective depth inference even when ray intersections are sparser and noisier, as is typical in the monocular setting. Apologies if the explanation was not clear.
>
> ## **Sensitivity to pose noise and drift from event-based SLAM system**:
>
> While it is true that our method depends on input poses, other 3D reconstruction pipelines like NeRFs and 3D Gaussian Splats also have the same assumption. Similar to them, our camera poses can be estimated using a parallel module.
> We show that even when we significantly reduce the quality of input poses by using other event-based SLAM systems, our depth estimation performance remains remarkebly robust.
>
> Our experimental pipeline: Stereo events --> estimated poses using event SLAM --> DSI --> Depth maps using DERD-Net
>
> (a) On DSEC zurich_city_04_a sequence, instead of GT poses (from LiDAR-IMU odometry), we used camera poses estimated by a stereo event odometry system ES-PTAM [1], which reported 131.62 cm Absolute Trajectory Error (ATE) in 50 m depth range scene, to run DERD-Net and achieved the following depth estimation performance:
>
> | Algorithm | Mean Err [m] | Median Err [m] | bad-pix [%] | # Points [million] |
> |---|---|---|---|---|
> | Noisy DERD-Net + F_orig        | 1.56 (-3.11%) | 0.45 (-2.17%)  | 3.84 (-6.8%)   | 1.61 (-3.59%) |
> | Noisy DERD-Net + F_denser   | 1.74 (-3.33%) | 0.52 (-3.7%)  | 4.84 (-3.97%)    | 4.49  (-3.23%) |
>
> Compared to Table 3, while the number of points for which depth is evaluated slighthly decreased, DERD-Net's depth estimation errors actually even improved. This demonstrates strong robustness to noisy poses obtained from an event SLAM system.
>
> (b) On MVSEC flying1, instead of GT poses (from motion capture), we used camera poses estimated by ES-PTAM [1] (ATE 14.93 cm in 6 m depth range scene) to run DERD-Net and achieved the following depth estimation performance:
>
> | Algorithm | Mean Err [cm] | Median Err [cm] | bad-pix [%] | # Points [million] |
> |---|---|---|---|---|
> | Noisy DERD-Net + F_orig        | 12.72 (+8.81%) | 6.33 (+16.79%)  | 0.65 (-24.42%)   | 0.68 (-30.61%) |
> | Noisy DERD-Net + F_denser   | 15.76 (+6.06%) | 7.36 (+13.76%)  | 1.17 (-10.69%)   | 1.62 (-46.18%) |
> | Noisy Multi-Pixel DERD-Net + F_orig   | 13.53 (+10.27%) |  6.76 (+19.01%) | 0.65 (-24.42%) | 3.41(-33.91%) |
> | Noisy Multi-Pixel DERD-Net + F_denser   | 16.60 (+6.62%) | 7.65 (+16.08%)  |  1.21 (-11.68%) | 6.27 (-46.46%) |
>
> Compared to Table 8, the MAE and MedAE worsened only slightly, while bad-pix even improved. The strongest decrease here was in the number of points. However, note that Noisy DERD-Net + F_denser still predicts depths on 69% more pixels than MC-EMVS + f_orig using ideal poses, while still yielding a 30% lower MAE. For its multi-pixel version, Noisy DERD-Net shows superior performance to all SOTA methods while predicting on the most points. This highlights again DERD-Net's strong robustness to noisy poses obtained from an event SLAM system.
>
> (c)  For completeness, we also used poses from SOTA event-based stereo visual-inertial odometry system ESVO2 [2] to run DERD-Net on MVSEC flying1, flying2 and flying3 and report mean metrics. Compared to ES-PTAM, ESVO2 is notably more accurate due to the visual-inertial data fusion, producing less noisy poses, which yield an average ATE of ~8.3 cm in 6 m depth range scene.
>
> | Algorithm | Mean Err [cm] | Median Err [cm] | bad-pix [%] | # Points [million] |
> |---|---|---|---|---|
> | Noisy DERD-Net + F_orig        | 11.53 (-1.37%) | 5.73 (+4.18%)  | 1.09 (+22.47%)   | 0.61 (-22.78%) |
> | Noisy DERD-Net + F_denser   | 13.69 (-10.17%) | 6.32 (-5.39%)  | 1.46 (-14.12%)   | 1.45 (-47.65%) |
>
> Compared to Table 3, this shows only a slight decrease in performance for DERD-Net's depth estimation error on f_orig, while it even improves on f_denser, thus indicating robustness to noisy poses obtained from a SLAM system using events and IMU. With regards to the decreased number of pixels, the same observations apply as for the table above.
>
> ## **Runtime analysis and comparison with SOTA**
> On a PC with Intel Xeon(R) W-2225 CPU operating at 4.10GHz with 8 cores, for MVSEC (DAVIS cameras with 346x260 pixels, 100 depth planes, 2 million events), each DSI construction takes ~45 ms, DSI fusion (for stereo) takes ~26 ms, and pixel selection takes ~0.2 ms. These values are common for both  DERD-Net and the SOTA method MC-EMVS.
>
> argmax for a single sub-DSI takes 0.02 ms, whereas DERD-Net inference for a single sub-DSI takes 0.37 ms. By parallelly processing sub-DSIs in GPU, this amounts to a total time of 0.06 ms for argmax and 1.12 ms for DERD-Net inference.
>
> As rightly pointed out, the time taken by DERD-Net inference is very small compared to DSI creation time. Using a network instead of argmax thus does not create a bottleneck in this framework.
>
> It has been shown that DSI creation does not hamper real-time performance with DAVIS cameras [2, 3] because the 3D map can be updated infrequently and on-demand. With an additional ~1 ms increase in total runtime, our method DERD-Net can also run real-time.
>
> # Answers to Questions:
> 1. Please refer to our explanation above about the monocular setting.
>
> 2. Please refer to the section above on sensitivity to noisy poses. For further experiments showing robustness, please see our response to Reviewer PC8y.
>
> 3. Please see our section above on runtime analysis for DERD-Net as well as the SOTA stereo event SLAM method MC-EMVS [1]. We produce comparable (real-time) performance while significantly improving accuracy.
>
> 4. Please refer to Table 6 and Section A.2 in the Appendix which shows how the network performance changes with increasing sub-DSI and consequently network size. Depth accuracy improves as the size increases but at a slower rate than number of operations.
>
> 5. As shown in Section A.2 in our appendix, the performance of our network is not saturated yet. Training for a single epoch with sub-DSI size 9x9 already yielded better performance than multiple epochs on the 7x7 sub-DSI. Training more epochs and an even slightly larger network size would improve performance even further.
> However, we expect a tipping point, where increasing network size would incentivise it to learn to over-fit to a particular scenario because it considers a large spatial window and derives semantic context, as opposed to learning about local ray intersections. Moreover, greatly increasing the sub-DSI size would at some point lead to a significant reduction of training data (due to fewer sub-DSIs). Our approach deals with limited labeled training data (which is a prevalent issue in event vision) by turning each pixel into a training sample, therefore augmenting the available dataset. Generalization performance could be further improved by data augmentation methods such as mirroring, DSIs computed with noisy poses and various depth resolutions.
>
> References:
>
> [1] Ghosh et al. ES-PTAM: Event-Based Stereo Parallel Tracking and Mapping. In ECCVW 2024.
>
> [2] Niu et al. ESVO2: Direct Visual-Inertial Odometry With Stereo Event Cameras. IEEE T-RO 41 (2025), 2164–2183.

---

> > ### Comment · Reviewer_KbkG · 2025-08-05
> >
> > Thank you for the response. The clarification and additional details are satisfactory. I will maintain the positive rating. Thanks.

---

> > > ### Author Response · Authors · 2025-08-07
> > >
> > > Thank you for your response and for acknowledging the clarifications and additional details we provided. We are glad to hear that the current version satisfactorily addresses your concerns.
> > >
> > > Is there anything we could address to support an improvement in rating?
> > >
> > > Thank you again for your time and constructive feedback.

---

> > > > ### Comment · Reviewer_KbkG · 2025-08-08
> > > >
> > > > I have updated the final rating to 5. Thank you for the effort in the response.

---

> > > > > ### Author Response · Authors · 2025-08-08
> > > > >
> > > > > Thank you for your reply and for updating the rating.

---

### Official Review · Reviewer_Yns5 · 2025-07-03

**Clarity:** 3
**Significance:** 3
**Originality:** 2
**Rating:** 4
**Confidence:** 3

**Summary:**

This paper proposes DERD-Net, a deep learning-based method for estimating depth from event camera data. The core methodology avoids direct processing of asynchronous event streams. Instead, it leverages the Disparity Space Image (DSI) representation, which is constructed by back-projecting events into a volumetric grid using known camera poses, a technique established in EMVS [9] and MC-EMVS [10]. The main contribution is a lightweight neural network that operates on small, local patches of the DS to predict pixel-wise depth. The network uses a combination of 3D convolutions and a GRU to interpret these local ray density patterns. The authors present experiments on the MVSEC and DSEC datasets, claiming state-of-the-art performance in both monocular and stereo settings, significant improvements in depth map completeness, and superior robustness compared to existing methods.

**Questions:**

- The performance of your method is contingent on the quality of input poses. How does DERD-Net's performance degrade as a function of increasing noise in the camera poses, simulating a realistic SLAM/VIO scenario? A sensitivity analysis is critical for understanding the practical utility of this work.

- Can you justify the comparison in Table 4 against end-to-end methods that do not require camera poses as input? Given the vastly different problem setups, how can any meaningful conclusion about comparative robustness be drawn?

- Could you provide a full pipeline timing analysis, including the DSI construction time for an average frame in the MVSEC dataset? This would give a more realistic picture of the method's computational cost compared to the reported sub-millisecond inference time.

**Ethical Concerns:**

["NO or VERY MINOR ethics concerns only"]

**Final Justification:**

Most of my major concerns have been addressed. The new evidence has convinced me of the value and robustness of the proposed contribution. I will increase my rating.

**Limitations:**

The authors have listed several limitations, such as the requirement for ground truth depth data for training and the need for input camera poses.

**Quality:**

2

**Strengths And Weaknesses:**

Strengths:
+ The paper is well-written and easy to follow.

+ The idea of applying a learned function to interpret the rich information in a DSI, as opposed to a simple argmax heuristic, is a sensible and logical next step for this line of research. The local processing on Sub-DSIs is an intelligent design choice for keeping the network lightweight and potentially improving generalization.

+ The authors have conducted experiments across two standard datasets, comparing their method against numerous baselines on ten different metrics.

Weaknesses:

- The foundational components—using event cameras for multi-view stereo, leveraging known camera poses, and the DSI data structure itself—are directly adopted from prior work, most notably EMVS [9] and MC-EMVS [10]. The proposed method, DERD-Net, essentially replaces the final argmax step of MC-EMVS with a neural network, which predictably yields better results than a non-parametric heuristic. The claim of being the "first learning-based multi-view stereo method to use camera poses along with events as input" is narrowly scoped and misleading, as the use of poses to create the geometric DSI representation is the main contribution of the prior work it builds upon.

- The entire experimental validation rests on the assumption: the availability of high-quality camera poses. The DSI construction is acutely sensitive to the accuracy of the input camera poses. In any real-world application, such as SLAM, these poses would be estimated and contain noise. The paper provides no analysis of the method's sensitivity to pose error. It is highly probable that the performance of DERD-Net would degrade dramatically with the noisy poses from a real tracking system. Moreover, this dependency makes the comparisons, particularly in Table 4, invalid. DERD-Net is compared against end-to-end stereo methods that do not require prior camera poses as input. These are methods tackling a harder, more complete problem. The authors concede that "absolute accuracy is not directly comparable", yet they proceed to make strong claims about superior robustness based on this comparison. This is not scientifically rigorous. The observed "robustness" is more likely a property of the DSI representation itself (which integrates information over time using clean poses) than a unique feature of the DERD-Net architecture.

- The paper emphasizes an "ultra-fast" inference time of 0.37 ms.  However, this is the time to process a single Sub-DSI. The true computational bottleneck is the construction of the DSI itself, which requires iterating through and back-projecting millions of events for each depth map. The time required for this crucial pre-processing step is not reported, leading to an incomplete and overly optimistic assessment of the method's efficiency for real-time applications.

---

> ### Author Rebuttal · Authors · 2025-07-30
>
> Thanks for your insightful comments, especially for appreciating the soundness of our approach and comprehensive evaluation.
>
> ## **On being the first learning-based method for multi-view stereo**
> From the perspective of input (events and poses) to output (depth map), our method is the first one to successfully leverage deep learning to provide high-performance depth estimation.
> Apologies for the lack of clarity regarding our scope.
> Although DSIs for event cameras were first proposed in EMVS [9] (IJCV 2018), the concept of DSI itself is based on space-sweeping [3] which was published in 1996.
> DSIs are simply an intermediate representation of our entire framework, chosen for its strong rich geometric encoding and multi-camera scalability.
>
> Even though the event-based DSI framework was proposed in 2018, there have been no successful attempts in combining it with machine learning. Learning from DSIs without introducing additional computational overheads has substantial technical challenges since directly processing the entire DSI volume would make the network too big for on-device and real-time applications, and would also be prone to overfitting due to the limited availability of training data.
> Since the DSI is mostly empty, a full-frame model would face a high-dimensional input space with sparse supervision, making it highly susceptible to overfitting and poor generalization.
> Moreover, a network consisting of only convolutions would not be able to model the sequential nature in which projected rays pass through space when creating the DSIs, which is needed to tackle variable depth resolution.
>
> Our key contribution lies in the additional step of extracting small independent sub-regions (Sub-DSIs) around relevant pixels that can be processed in parallel.
> This novel design takes advantage of the inherent sparsity of event data and the DSI to overcome the aforementioned challenges and provide the following advantages instead: (i) independence from camera resolution; (ii) enabling of parallel processing; (iii) extension of the otherwise limited training set (e.g. for MVSEC the dataset increases from a few thousand to a more than a million data points); (iv) yields an ultra-lightweight network with very low constant memory occupation and inference time; (v) enhanced robustness by preventing the network from remembering the scene.
>
> To the best of our knowledge, this work is also the first learning-based  work to tackle "non-instantaneous" stereo depth estimation where long-term temporal aggregation is need from multiple viewpoints. With our approach, we argue that extracting camera pose information first and then explicitly using them for depth estimation yields better results than end-to-end learning methods (e.g., Table 4).
>
> ## **Sensitivity to pose noise and drift from event-based SLAM system**:
>
> While DERD-Net relies on input camera poses, they do not need to be of "high quality".
> Availability of camera poses is a common assumption in many 3D reconstruction pipelines, including popular methods like NeRF and 3D Gaussian Splatting.
> Following the reviewers suggestion, we ran experiments to investigate DERD-Net's performance with noisy camera poses.
> Contrary to the concern raised, our results show that DERD-Net remains robust even when the input poses are degraded by noise and drift from a SLAM system.
> Below, we report results using input poses estimated by event-based SLAM pipelines (ES-PTAM and ESVO2).
>
> Our experimental pipeline: Stereo events --> estimated poses using event SLAM --> DSI --> Depth maps using DERD-Net.
>
> (a) On DSEC zurich_city_04_a sequence, instead of GT poses (from LiDAR-IMU odometry), we used camera poses estimated by a stereo event odometry system ES-PTAM [1], which reported 131.62 cm Absolute Trajectory Error (ATE) in 50 m depth range scene, to run DERD-Net and achieved the following depth estimation performance:
>
> | Algorithm | Mean Err [m] ↓ | Median Err [m] ↓ | bad-pix [%] ↓ | # Points [million] ↑ |
> |---|---|---|---|---|
> | Noisy DERD-Net + F_orig        | 1.56 (-3.11%) | 0.45 (-2.17%)  | 3.84 (-6.8%)   | 1.61 (-3.59%) |
> | Noisy DERD-Net + F_denser   | 1.74 (-3.33%) | 0.52 (-3.7%)  | 4.84 (-3.97%)   | 4.49 (-3.23%) |
>
> Compared to Table 3, while the number of points for which depth is evaluated slightly decreased, DERD-Net's depth estimation errors actually even improved. This demonstrates strong robustness to noisy poses obtained from an event SLAM system.
>
> (b) On MVSEC flying1, instead of GT poses (from motion capture), we used camera poses estimated by ES-PTAM [1] (ATE 14.93 cm in 6 m depth range scene) to run DERD-Net and achieved the following depth estimation performance:
>
> | Algorithm | Mean Err [cm] ↓ | Median Err [cm] ↓ | bad-pix [%] ↓ | # Points [million] ↑ |
> |---|---|---|---|---|
> | Noisy DERD-Net + F_orig        | 12.72 (+8.81%) | 6.33 (+16.79%)  | 0.65 (-24.42%)   | 0.68 (-30.61%) |
> | Noisy DERD-Net + F_denser   | 15.76 (+6.06%) | 7.36 (+13.76%)  | 1.17 (-10.69%)   | 1.62 (-46.18%) |
> | Noisy Multi-Pixel DERD-Net + F_orig   | 13.53 (+10.27%) |  6.76 (+19.01%) | 0.65 (-24.42%) | 3.41(-33.91%) |
> | Noisy Multi-Pixel DERD-Net + F_denser   | 16.60 (+6.62%) | 7.65 (+16.08%)  |  1.21 (-11.68%) | 6.27 (-46.46%) |
>
> Compared to Table 8, the MAE and MedAE worsened only slightly, while bad-pix even improved. The strongest decrease here was in the number of points. However, note that Noisy DERD-Net + F_denser still predicts depths on 69% more pixels than MC-EMVS + F_orig using ideal poses, while still yielding a 30% lower MAE. For its multi-pixel version, Noisy DERD-Net shows superior performance to all SOTA methods while predicting on the most points. This highlights again DERD-Net's strong robustness to noisy poses obtained from an event SLAM system.
>
> (c)  For completeness, we also used poses from SOTA event-based stereo visual-inertial odometry system ESVO2 [2] to run DERD-Net on MVSEC flying1, flying2 and flying3 and report mean metrics. Compared to ES-PTAM, ESVO2 is notably more accurate due to the visual-inertial data fusion, producing less noisy poses, which yield an average ATE of ~8.3 cm in 6 m depth range scene.
>
> | Algorithm | Mean Err [cm] ↓ | Median Err [cm] ↓ | bad-pix [%] ↓ | # Points [million] ↑ |
> |---|---|---|---|---|
> | Noisy DERD-Net + F_orig        | 11.53 (-1.37%) | 5.73 (+4.18%)  | 1.09 (+22.47%)   | 0.61 (-22.78%) |
> | Noisy DERD-Net + F_denser   | 13.69 (-10.17%) | 6.32 (-5.39%)  | 1.46 (-14.12%)   | 1.45 (-47.65%) |
>
> Compared to Table 3, this shows only a slight decrease in performance for DERD-Net's depth estimation error on F_orig, while it even improves on F_denser, thus indicating robustness to noisy poses obtained from a SLAM system using events and IMU.
>
> In all cases, depth estimation performance remains nearly unchanged.
>
> ## **Runtime analysis**
> On a PC with Intel Xeon(R) W-2225 CPU operating at 4.10GHz with 8 cores, for MVSEC (DAVIS cameras with 346x260 pixels, 100 depth planes, 2 million events), each DSI construction takes ~45 ms, DSI fusion (for stereo) takes ~26 ms, and pixel selection takes ~0.2 ms. These values are common for both the SOTA method MC-EMVS and DERD-Net.
>
> argmax for a single sub-DSI takes 0.02 ms, whereas DERD-Net inference for a single sub-DSI takes 0.37 ms. By parallelly processing sub-DSIs in GPU, this amounts to a total time of 0.06 ms for argmax and 1.12 ms for DERD-Net inference.
>
> As rightly pointed out, the time taken by DERD-Net inference is very small compared to DSI creation time. Using a network instead of argmax thus does not create a bottleneck in this framework.
>
> It has been shown that DSI creation does not hamper real-time performance with DAVIS cameras [4, 5] because the 3D map can be updated infrequently and on-demand. With an additional ~1 ms increase in total runtime, our method DERD-Net can also run real-time.
>
> # Answers to Questions:
> 1. Please refer to our sensitivity analysis to noisy poses above. We would also refer to additional experiments done for checking robustness in our response to Reviewer PC8y.
>
> 2. To the best of our knowledge, our method is the first to provide a learning-based solution to the problem of long-term depth estimation from event data, which needs ego-motion information to properly fuse event data over time (i.e., multi-view). Other learning-based methods in the literature (Table 4) are short-term (in the style of MegaDepth's single-view depth prediction), hence they do not use or leverage ego-motion information. While the setups are different, we analyze how precise the value are. In Lines 313-319, we provide a caveat against blind comparison.
> Each method in Table 4 produces three accuracy numbers (according to the data splits); we compare how similar these numbers are per method (i.e., precision). Example: our method has ratios 11.11/11.69=0.95 and 12.28/11.69=1.05 with respect to split 1, whereas other methods have larger dissimilarities (i.e., lower precision), e.g., EIS (ICCV 21): 18.43/13.74=1.34 and 22.36/13.74=1.62. That is, we use precision or "spread" of the results as a measure of robustness.
>
> 3. Please refer to our runtime analysis above.
>
> References:
>
> [1] Ghosh et al. ES-PTAM: Event-Based Stereo Parallel Tracking and Mapping. In ECCVW 2024.
>
> [2] Niu et al. ESVO2: Direct Visual-Inertial Odometry With Stereo Event Cameras. IEEE T-RO 41 (2025), 2164–2183.
>
> [3] R. T. Collins, "A space-sweep approach to true multi-image matching," CVPR 1996, doi: 10.1109/CVPR.1996.517097.
>
> [4] Ghosh, Gallego. "Live 3D reconstruction using stereo event cameras", CVPR 2023 demo.
>
> [5] Ghosh, Cavinato, Gallego. SLAM with Stereo Event Cameras. ECCV 2024 demo.

---

> > ### Author Response · Authors · 2025-08-07
> > **Request to Initialize Discussion**
> >
> > Thank you again for your review. As the discussion period closes, we would greatly appreciate it if you could share any remaining concerns or points of clarification you might have, so we can address them while there is still time.

---

> > ### Comment · Reviewer_Yns5 · 2025-08-09
> >
> > Thanks for the rebuttal. Most of my major concerns have been addressed. The new evidence has convinced me of the value and robustness of the proposed contribution. I will increase my rating.

---

> > > ### Author Response · Authors · 2025-08-09
> > >
> > > Thank you for your comment and for updating the rating. We are glad we could address most of your major concerns and that the new evidence has helped convey the value and robustness of our contribution. We also appreciate your role in helping us improve the work.

---

### Official Review · Reviewer_PC8y · 2025-07-04

**Clarity:** 3
**Significance:** 3
**Originality:** 3
**Rating:** 4
**Confidence:** 4

**Summary:**

This paper presents DERD-Net, a novel method for depth estimation from event cameras using the spatial density of back-projected event rays, represented as Disparity Space Images (DSIs). The proposed framework incorporates known camera poses to generate DSIs and processes local subregions (Sub-DSIs) with a lightweight neural network combining 3D convolutions and a gated recurrent unit (GRU). By focusing on small volumetric neighborhoods, the approach ensures constant model complexity, efficient parallel inference, and resolution independence. DERD-Net supports both monocular and stereo configurations and does not require event synchronization across views. Experiments on the MVSEC and DSEC datasets demonstrate that DERD-Net achieves superior performance compared to existing methods across multiple metrics, including mean and median error, depth completeness, and robustness. The framework also generalizes well across different scenes and modalities, and its efficient design makes it suitable for applications requiring real-time event-based depth estimation.

**Questions:**

1. The proposed method relies on externally provided camera poses to construct DSIs. Could the authors elaborate on the method's sensitivity to pose noise or drift, and whether any tolerance analysis has been performed?

2. Given the method's stated motivation for SLAM applications, would the authors consider including evaluations on downstream tasks such as visual odometry or mapping to substantiate its practical impact?

**Ethical Concerns:**

["NO or VERY MINOR ethics concerns only"]

**Final Justification:**

The authors have addressed my concerns raised during the review process.

**Limitations:**

The paper assumes the availability of accurate camera poses but does not analyze the method's robustness to pose estimation errors. This reliance may affect the applicability of the approach in real-world scenarios where pose inputs are noisy or unreliable.

**Quality:**

3

**Strengths And Weaknesses:**

Strengths:

1. Novel methodology: The paper introduces DERD-Net, a unique depth estimation framework that leverages event-ray densities (DSIs), representing a shift from traditional event stream processing.

2. Efficient and compact design: The use of 3D convolutions and GRU on local Sub-DSIs enables accurate predictions with a lightweight model (~70k parameters) and sub-millisecond inference time, making it suitable for real-time applications.

3. Comprehensive evaluation: The method is thoroughly tested on MVSEC and DSEC datasets, showing consistent improvements over state-of-the-art methods across accuracy and completeness metrics.

Weakness:

1. Reliance on camera poses: The method depends on accurate external pose inputs for DSI construction, but the paper does not analyze how pose noise affects performance, which could impact robustness in practice.

---

> ### Author Rebuttal · Authors · 2025-07-30
>
> Thanks for your feedback reviewing our paper, especially for recognizing the novelty of our method, efficient design and comprehensive evaluation.
>
> Thanks to the suggestion, we performed additional experiments to measure robustness of our approach against imperfect (noisy) camera poses and downstream camera tracking performance. Our experimental results showed that DERD-Net remains remarkebly robust even when the input poses are significantly degraded, as would be the case in real-world SLAM scenarios:
>
> ## **Exp 1: Depth estimation using noisy poses (with drift) from event-based SLAM system**:
>
> Pipeline: Stereo events --> estimated poses using event SLAM --> DSI --> Depth maps using DERD-Net
>
> (a) On DSEC zurich_city_04_a sequence, instead of GT poses (from LiDAR-IMU odometry), we used camera poses estimated by a stereo event odometry system ES-PTAM [1], which reported 131.62 cm Absolute Trajectory Error (ATE) in 50 m depth range scene, to run DERD-Net and achieved the following depth estimation performance:
>
> | Algorithm | Mean Err [m] ↓ | Median Err [m] ↓ | bad-pix [%] ↓ | # Points [million] ↑ |
> |---|---|---|---|---|
> | Noisy DERD-Net + F_orig        | 1.56 (-3.11%) | 0.45 (-2.17%)  | 3.84 (-6.8%)   | 1.61 (-3.59%) |
> | Noisy DERD-Net + F_denser   | 1.74 (-3.33%) | 0.52 (-3.7%)  | 4.84 (-3.97%)    | 4.49  (-3.23%) |
>
> Compared to Table 3, while the number of points for which depth is evaluated slightly decreased, DERD-Net's depth estimation errors actually even improved. This demonstrates strong robustness to noisy poses obtained from an event SLAM system.
>
> (b) On MVSEC flying1, instead of GT poses (from motion capture), we used camera poses estimated by ES-PTAM [1] (ATE 14.93 cm in 6 m depth range scene) to run DERD-Net and achieved the following depth estimation performance:
>
> | Algorithm | Mean Err [cm] ↓ | Median Err [cm] ↓ | bad-pix [%] ↓ | # Points [million] ↑ |
> |---|---|---|---|---|
> | Noisy DERD-Net + F_orig        | 12.72 (+8.81%) | 6.33 (+16.79%)  | 0.65 (-24.42%)   | 0.68 (-30.61%) |
> | Noisy DERD-Net + F_denser   | 15.76 (+6.06%) | 7.36 (+13.76%)  | 1.17 (-10.69%)   | 1.62 (-46.18%) |
> | Noisy Multi-Pixel DERD-Net + F_orig   | 13.53 (+10.27%) |  6.76 (+19.01%) | 0.65 (-24.42%) | 3.41(-33.91%) |
> | Noisy Multi-Pixel DERD-Net + F_denser   | 16.60 (+6.62%) | 7.65 (+16.08%)  |  1.21 (-11.68%) | 6.27 (-46.46%) |
>
> Compared to Table 8, the MAE and MedAE worsened only slightly, while bad-pix even improved. The strongest decrease here was in the number of points. However, note that Noisy DERD-Net + F_denser still predicts depths on 69% more pixels than MC-EMVS + F_orig using *ideal* poses, while still yielding a 30% lower MAE. For its multi-pixel version, Noisy DERD-Net shows superior performance to all SOTA methods while predicting on the most points. This again highlights DERD-Net's strong robustness to noisy poses obtained from an event SLAM system.
>
> (c)  For completeness, we also used poses from SOTA event-based stereo visual-inertial odometry system ESVO2 [2] to run DERD-Net on MVSEC flying1, flying2 and flying3 and report mean metrics. Compared to ES-PTAM, ESVO2 is notably more accurate due to the visual-inertial data fusion, producing less noisy poses, which yield an average ATE of ~8.3 cm in 6 m depth range scene.
>
> | Algorithm | Mean Err [cm] ↓ | Median Err [cm] ↓ | bad-pix [%] ↓ | # Points [million] ↑ |
> |---|---|---|---|---|
> | Noisy DERD-Net + F_orig        | 11.53 (-1.37%) | 5.73 (+4.18%)  | 1.09 (+22.47%)   | 0.61 (-22.78%) |
> | Noisy DERD-Net + F_denser   | 13.69 (-10.17%) | 6.32 (-5.39%)  | 1.46 (-14.12%)   | 1.45 (-47.65%) |
>
> Compared to Table 3, this shows only a slight decrease in performance for DERD-Net's depth estimation error on f_orig, while it even improves on f_denser, thus indicating robustness to noisy poses obtained from a SLAM system using events and IMU. We observe the same trend in the number of pixels as the table in (b).
>
> ## **Exp 2: Downstream camera tracking performance using depth estimated from DERD-Net**
>
> Pipeline: Stereo events + GT poses (from LiDAR and IMU) → DSI → Depth and point-cloud estimated using DERD-Net → Camera pose estimated by aligning 3D map to events
>
> We report camera tracking performance (Absolute Trajectory Errors (ATE) and Absolute Rotation Errors (ARE)) on all zurich_city_04 sequences of DSEC driving dataset, even though the DERD-Net model was trained only on the zurich_city_04_a sequence to highlight generalization capabilities.
>
> | Sequence | Duration [s] | ATE RMSE [cm] ↓ | ARE RMSE [deg] ↓ |
> |---|---|---|---|
> | zc04a  | 35 | 17.07 | 0.31  |
> | zc04b  |  13.4 | 7.85 | 0.08  |
> | zc04c  | 53 | 14.00 | 0.45  |
> | zc04d  | 47.8 | 55.64 | 0.67  |
> | zc04e  |  13.6 | 5.71 | 0.11  |
> | zc04f  | 43.1 | 36.11 | 0.72  |
>
> These camera pose errors in the 50 m depth range scene across all sequences shows strong performance of DERD-Net for downstream tasks (pose estimation using simple photometric edge alignment on event images), as well as robust generalization even when trained on a single sequence. Training on more diverse DSEC sequences would improve performance even further.
>
> ## **Exp 3: Closing the loop: Depth estimation performance on poses computed downstream of DERD-Net**
>
> Pipeline: Stereo events + GT poses (from LiDAR and IMU) → DSI → Depth and point-cloud estimated using DERD-Net → Camera pose estimated by aligning 3D map to events → DSI → Depth estimated using DERD-Net
>
> To go one step further and close the loop, we use the poses estimated above for DSEC zurich_city_04_a and re-run DERD-Net to measure robustness to noise generated by its own previous iteration.
>
> | Algorithm | Mean Err [m] ↓ | Median Err [m] ↓ | bad-pix [%] ↓ | # Points [million] ↑ |
> |---|---|---|---|---|
> | Noisy DERD-Net + F_orig        | 1.66 (+3.11%) | 0.49 (+6.52%)   | 4.19 (+1.7%)    | 1.46 (-12.57%) |
> | Noisy DERD-Net + F_denser   | 1.95 (+8.33%) | 0.60 (+11.11%)   |  5.65 (+12.1%)  | 4.09 (-11.85%) |
>
> Performance showed only minor degradation, particularly for f_orig, with no metric worsening by more than 13%. Remarkably, even under such noisy pose conditions, Noisy DERD-Net’s depth estimation errors remain around 50% better than those of prior SOTA methods using ideal poses. This demonstrates the practical viability of using DERD-Net as a depth estimation module within a self-sustaining SLAM system.
>
> # Answers to Questions:
> 1. Please refer to experiments 1 and 3 above for sensitivity analysis to pose noise and drift.
> 2. Please refer to experiments 2 and 3 above for downstream performance in camera tracking and re-mapping using estimated poses.
>
> References:
>
> [1] Ghosh et al. ES-PTAM: Event-Based Stereo Parallel Tracking and Mapping. In ECCVW 2024.
>
> [2] Niu et al. ESVO2: Direct Visual-Inertial Odometry With Stereo Event Cameras. IEEE T-RO 41 (2025), 2164–2183.

---

> > ### Comment · Reviewer_PC8y · 2025-08-06
> >
> > The authors have addressed my concerns.

---

> > > ### Author Response · Authors · 2025-08-07
> > >
> > > Thank you for your response. We’re glad to hear that the rebuttal resolves the issues you raised. Your thoughtful review contributed to improving the paper, which we greatly appreciate.
> > >
> > > Is there anything else we could address to support an improvement in rating?

---

### Note · Authors · 2025-08-15

We thank the reviewers (R1=PC8y, R2=Yns5, R3=KbkG, R4=gXBu) for their positive feedback and suggestions, which helped us further strengthen our work.

**Summary**

Pre-rebuttal, reviewers recognized the novelty and strong empirical results of our approach -- it achieves “consistent improvements over state-of-the-art methods” [R1] and “outperforms existing methods on standard benchmarks” [R3], with “very convincing metrics on multiple datasets” [R4].

They highlighted that our method “estimates depth in both monocular and stereo” [R4], “does not require event synchronization across views” [R1], and achieves “state-of-the-art performance in both monocular and stereo settings” [R2], with “surprisingly good depth estimates even from a single event camera” [R3].

Our “intelligent” [R2] and “efficient and compact design” [R1] enables “accurate predictions with a lightweight model (~70k parameters) and sub-millisecond inference time, making it suitable for real-time applications” [R1]. Thus, our “novel methodology” [R1] was found to be “low-latency and ideal for real-world applications” [R2] and “a unique depth estimation framework … representing a shift from traditional event stream processing” [R1].

**Main issues raised and addressed during rebuttal**

All reviewers encouraged us to demonstrate further robustness of our method to imperfect (noisy) camera poses. In response, we conducted extensive experiments to show robustness to noise and drift in poses from a SLAM system (see R1 rebuttal).
Reviewers also asked for detailed runtime analysis, which we provided during the rebuttal, showing that our method can run real-time (see R2 rebuttal).
For viability in a SLAM system, we also demonstrated downstream camera tracking performance, and re-mapping using the estimated camera poses, thereby closing the loop (see R1 rebuttal).
These additional results and analysis improve the completeness and quality of the paper and will be included in the final version.

**Outcome of rebuttal and discussion**

*All* reviewers expressed satisfaction with our rebuttal and explicitly confirmed their concerns have been addressed. Upon follow-up, no further concerns were raised. The additional experiments convinced the reviewers about the quality of our work, leading to an increase in rating in at least 2 cases; although R1 expressed that all concerns were addressed, we don't know if the rating 4 was increased. In summary, the reviews reflect a positive consensus post-rebuttal.

---

### Decision · Program_Chairs · 2025-09-17

**Decision:**

Accept (spotlight)

**Comment:**

The proposes a novel paradigm for estimating sparse depth maps from event cameras by introducing Disparity Space Images (DSIs) for events. The method achieves state-of-the-art results (PC8y, Yns5) in both monocular and stero settings, with an extremely low latency time of 1ms.

The paper received two borderline and two strong accepts. The AC agrees with these reviews and recommends the paper for acceptance.